# Integrative bioinformatics and experiments identify *RIBC2* as a key regulator in the esophageal cancer

Xuan Zheng[1], Yishuang Cui[2], Xuemin Yao[2], Yanan Wu[1], Yanlei Ge[2,3], Ye Jin[4,5], Junqing Gan[2,6], Weinan Yao[2,3], Yanna Bi[2,3], Guogui Sun [2,5]*

1 School of Public Health, North China University of Science and Technology, Tangshan, Hebei, China, 2 Affiliated Hospital of North China University of Science and Technology, Tangshan, Hebei, China, 3 Hebei Province Innovation Center for Biological Cell Function Development and Precise Detection Technology, Tangshan, Hebei, China, 4 Clinical Medicine School, North China University of Science and Technology, Tangshan, Hebei, China, 5 Department of Hebei Key Laboratory of Medical-Industrial Integration Precision Medicine, Tangshan, Hebei, China 6 Sun Guogui Innovation Studio, Tangshan, Hebei, China

* guogui_sun2021@sina.com

## Abstract

Early detection of esophageal cancer (EC) remains a major challenge due to the limited understanding of its initial molecular alterations. Therefore, this study aimed to identify the key molecular drivers involved in EC carcinogenesis. Human normal esophageal epithelial cells were subjected to chronic malignant transformation, followed by assessment of their morphological changes, proliferative capacity, clonogenic potential, migration, and invasion abilities. To elucidate the molecular mechanisms underlying tumorigenesis, transcriptome sequencing was performed and integrated with clinical datasets from two independent EC cohorts. Machine learning algorithms were then applied to pinpoint diagnostic and prognostic gene signatures, which were further validated through comprehensive in vitro and in vivo experiments. Differential expression analysis and machine learning identified RIB43A domain with coiled-coils 2 (RIBC2) as a strong diagnostic and prognostic biomarker for EC. RIBC2 expression was markedly upregulated in chronically transformed epithelial cells, established EC cell lines, and clinical tumor specimens, and its elevation was associated with unfavorable clinicopathological characteristics. Functional studies revealed that silencing RIBC2 significantly inhibited cell proliferation, migration, and invasion in both transformed and EC cells. Moreover, immune profiling indicated that high RIBC2 expression was linked to an immune-excluded tumor microenvironment, implying a potential role in modulating responsiveness to immunotherapy. These findings reveal RIBC2 as a novel driver of EC initiation and progression, highlighting its potential as a biomarker for early diagnosis and as a promising target for therapeutic intervention.

**Data availability statement:** All relevant data are within the paper and its Supporting information files. Raw sequencing data are available in the NCBI Sequence Read Archive (SRA) under BioProject accession PRJNA1391879 (ID: 1391879) (https://www.ncbi.nlm.nih.gov/bioproject/PRJNA1391879).

**Funding:** This research was funded by the National Natural Science Foundation of China, grant number 82472636; Hebei Province Innovation Capability Enhancement Plan Project, grant number 235A2403D; Hebei Provincial Department of Education Hebei Experimental Teaching and Teaching Laboratory Construction Project, grant number 81 and High level Research and Innovation Team Construction Plan of School of Public Health, North China University of Science and Technology, grant number KYTD202309. The funders had no role in study design, data collection and analysis, decision to publish, or preparation of the manuscript.

**Competing interests:** The authors have declared that no competing interests exist.

## Introduction

Esophageal cancer (EC) is the eighth most prevalent malignancy globally and the sixth leading cause of cancer-related deaths (https://gco.iarc.who.int/media/globocan/factsheets/populations/900-world-fact-sheet.pdf). Notably, nearly 52.7% of all EC-related fatalities worldwide occur in China [1]. Despite advances in first-line treatment options—including surgery, radiotherapy, chemotherapy, targeted therapies, and novel immunotherapeutic approaches, the overall prognosis for EC patients remains dismal. This is primarily attributed to challenges in early detection and the high recurrence rate after treatment [2]. These challenges underscore an urgent need to identify novel and robust biomarkers capable of facilitating early diagnosis and timely therapeutic intervention, ultimately improving patient outcomes in EC.

With the advent of high-throughput sequencing and integrative multi-omics technologies, bioinformatics has emerged as a powerful tool for cancer biomarker discovery [3,4]. Through the systematic analysis of large-scale datasets, bioinformatic approaches enable the identification of critical molecular alterations and signaling pathways involved in tumor initiation and progression. In the case of EC, computational analyses have proven valuable for prioritizing candidate biomarkers with diagnostic, prognostic, and therapeutic potential. For instance, Ren et al. utilized transcriptomic data from the TCGA-EC cohort to identify *ANGPTL7*, *CSRP1*, *EXPH5*, *F2RL2*, *KCNMA1*, *MAGEC3*, *MAMDC2*, and *SLC4A9* as fibroblast-associated gene signatures linked to EC prognosis [5]. Similarly, Yuan et al. analyzed TCGA gene expression profiles and identified miR-205-3p, miR-452-3p, and miR-6499-3p as promising biomarkers for EC staging [6]. By high-dimensional single-cell proteomics, Han et al. built a CCR4/CCR6 chemokine-based model to stratify EC patients with different response to neoadjuvant chemoradiotherapy combined with immunotherapy [7]. The microbiomes of EC patients showed that intratumoral Streptococcus signatures could predict the response of neoadjuvant chemoradiotherapy [8]. Additionally, integrative multi-omics, combining transcriptomics, proteomics, metabolomics and microbiomics may provide complementary signals and better understanding of molecular mechanisms [9]. In EC, for example, PGK1 is found to reprogram glucose metabolism and contributes to EC progression by analyzing data of genomics, proteomics and phosphoproteomics from EC patients covering 9 histopathological stages and 3 phases [10]. Integrated analyses of genomics, epigenomics, transcriptomics and proteomics could classify EC patients into four molecular subtypes and identify 28 features in the prediction of anti-PD1 therapy response [11].

Machine learning provides practical tools for feature selection and model construction when analyzing multi-omics data, and may improve the diagnostic accuracy and prognostic prediction in EC. For example, HSPD1 and MAP1LC3B were identified as key prognostic genes by LASSO regression, random forest and XGBoost algorithms [12]. Li et al. established a clinic-radiomics nomogram by machine learning to predict the overall survival after definitive chemotherapy of EC patients [13]. Sun et al. employed 10 machine learning algorithms to generate 101 algorithm combinations and identified key neoantigen-related genes. These genes could stratify EC patients into groups with differential infiltration levels of dendritic cells, macrophages and B

cells [14]. However, the strength of bioinformatics predictions is often constrained by the lack of clinical and experimental validation. Therefore, transforming in silico findings into clinically meaningful insights demands rigorous analytical frameworks coupled with comprehensive biological verification.

To overcome the aforementioned limitations, we developed a comprehensive experimental and computational framework to uncover key molecular drivers of EC. Specifically, we established an in vitro EC transformation model and generated corresponding gene expression profiles to capture the molecular alterations associated with malignant progression. These experimental datasets were subsequently integrated with transcriptomic data from the TCGA-EC cohort to enhance the robustness and clinical relevance of biomarker discovery. Furthermore, in vitro and in vivo assays were conducted to experimentally validate the functional role of the candidate markers in EC progression. Collectively, this study provides a systematic and experimentally grounded strategy for elucidating the molecular mechanisms underlying esophageal tumorigenesis. The insights gained from this work may pave the way for the development of novel biomarkers and therapeutic targets, ultimately improving the early detection, prognosis, and clinical management of EC.

## Materials and methods

### Tissue samples and cell lines

A total of 50 paired tissue samples, comprising EC tissues and their corresponding adjacent non-tumorous tissues, were collected from the Pathology Department of Tangshan People's Hospital. The adjacent tissues, located at least 5.0 cm from the tumor margin, were verified as histologically normal through hematoxylin and eosin (H&E) staining. All experiments involving human tissues and clinical information were approved by the Ethics Committee of Tangshan People's Hospital (Approval No.: RMYY-LLKS-2024042). All experiments were performed in accordance with the Declaration of Helsinki. Informed consents were obtained from all participants in the current study.

The human normal esophageal epithelial cell line Het-1A (RRID: CVCL_3702, male) was obtained from IMMOCELLBio (Xiamen, China) and maintained in Endothelial Cell Medium (Cat. No. 1001, ScienCell). The human EC cell lines KYSE-410 (esophageal squamous cell carcinoma, ESCC; RRID: CVCL_1352, male), TE-10 (ESCC; RRID: CVCL_1760, male), and OE33 (esophageal adenocarcinoma, EAC; RRID: CVCL_0471, female) were purchased from ProcellBio (Wuhan, China). These EC cell lines were cultured in RPMI-1640 medium (Cat. No. C11875500BT, Gibco) supplemented with 10% fetal bovine serum (FBS; Cat. No. 10099−141, Gibco) and 1% penicillin–streptomycin (Cat. No. 15140−122, Gibco). For cell passaging or subculturing, adherent cells were detached using 0.25% trypsin–EDTA (Cat. No. 25200−056, Gibco). All cells were maintained at 37°C in a humidified incubator containing 5% $CO_2$. The identity of all cell lines was verified by short tandem repeat (STR) profiling within the past three years, and all experiments were carried out using mycoplasma-free cultures.

### N-methyl-N′-nitro-N-nitrosoguanidine (MNNG) exposure and malignant transformation

After the Het-1A cells had fully adhered, they were divided into two experimental groups: Het-1A-T (treated with MNNG) and Het-1A-N (treated with solvent control). MNNG was first dissolved in DMSO to prepare a 1 mol/L stock solution, which was subsequently diluted with culture medium to achieve a final concentration of 2 μM. When the cells reached 60–70% confluency, they were exposed to 2 μM MNNG or the corresponding solvent control for 24 hours per passage, and this treatment cycle was repeated for 50 consecutive generations. The selected environmental exposure dose of MNNG was based on previously published studies.

### Data collection and processing

Two independent cohorts from the TCGA and GEO databases were used in this study to identify key genes associated with the diagnosis and prognosis of EC. The TCGA-EC cohort comprised 161 EC tissues and 11 normal esophageal tissues, while the GEO-EC dataset (GSE53625) included 179 EC samples and 179 matched normal tissues. The

corresponding transcriptomic profiles and clinical data from both cohorts were retrieved for comprehensive integrative analysis.

In parallel, total RNA was extracted from three Het-1A-T samples (MNNG-treated) and three Het-1A-N samples (solvent-treated controls) using the mirVana™ miRNA Isolation Kit (Cat. No. AM1561, Ambion) according to the manufacturer's instructions. RNA integrity was assessed using the Agilent 2100 Bioanalyzer (Agilent Technologies, Santa Clara, CA, USA), and only samples with an RNA Integrity Number (RIN) ≥ 7 were included for subsequent analyses. RNA sequencing libraries were prepared using the TruSeq Stranded Total RNA with Ribo-Zero Gold Kit (Illumina) following standard protocols, and sequencing was performed on the Illumina HiSeq™ 2500 platform (or equivalent), generating 125 bp/150 bp paired-end reads. The raw sequencing data were obtained in FASTQ format and underwent quality control filtering to remove low-quality reads and adapter sequences. The resulting high-quality clean reads were then aligned to the human reference genome for downstream transcriptomic analysis.

All sequencing data were normalized and expressed as fragments per kilobase of transcript per million mapped reads (FPKM). Differentially expressed genes (DEGs) were identified using the DESeq2 package, with comparisons made between EC and normal tissues (DEGs1) and between Het-1A-T and Het-1A-N groups (DEGs2). Genes meeting the criteria of $|\log_2$ fold change$| > 1$ and an adjusted p-value $< 0.05$ were considered statistically significant.

## Bioinformatics analyses

The overlapping DEGs derived from DEGs1 and DEGs2 were first subjected to univariate Cox regression analysis to identify genes significantly associated with EC prognosis. The functional enrichment of these key DEGs was performed using ClueGO to elucidate their potential biological roles. Subsequently, the key DEGs were analyzed using four machine learning algorithms, CatBoost, Random Forest, Support Vector Machine (SVM), and Logistic Regression to construct a robust predictive model for EC occurrence. In parallel, Least Absolute Shrinkage and Selection Operator (LASSO) regression was applied to select feature genes from the key DEGs, which were then used to calculate a risk score. The association between the risk score and patient survival was evaluated through Kaplan–Meier (K–M) survival analysis and receiver operating characteristic (ROC) curve assessment. Furthermore, the biological functions and immune landscape associated with the identified feature genes were explored by comparing the high-risk and low-risk groups, providing insights into their roles in EC progression and tumor immune microenvironment modulation. The basic information of EC patients in low- and high-risk groups were shown in S1 Table.

In addition, to investigate the role of RIBC2 in tumor immune microenvironment and its potential molecular mechanisms in regulating EC, we (1) estimated the immune infiltration by IOBR package [15], (2) performed GSEA by GseaVis package [16], (3) identified the DEGs by DESeq2 using $|\log_2$ fold change$| > 1$ and an adjusted p-value $< 0.05$, and performed functional enrichment and protein-protein interaction network in low- and high-RIBC2 expression groups.

## RNA extraction and reverse transcription quantitative polymerase chain reaction (RT-qPCR)

Total RNA was isolated from paraffin-embedded tissue samples and cell cultures using the Paraffin-Embedded Tissue Total RNA Extraction Kit (Cat. No. R310, GeneBetter) and the Universal Total RNA Kit (Cat. No. R013, GeneBetter), respectively, in accordance with the manufacturers' protocols. The concentration and purity of the extracted RNA were quantified using a SpectraMax QuickDrop spectrophotometer (Molecular Devices, USA). Subsequently, complementary DNA (cDNA) was synthesized from the RNA samples using the PrimeScript™ RT Master Mix (Cat. No. RR036A, TaKaRa) following the manufacturer's instructions. Quantitative real-time PCR (qPCR) was performed on a QuantStudio™ 3 System (Thermo Fisher Scientific, USA) employing TB Green® Premix Ex Taq™ II (Cat. No. RR820A, TaKaRa). For the detection of RIBC2 expression, qPCR was carried out under the following cycling conditions: initial denaturation at 95°C for 30 seconds, followed by 40 cycles of denaturation at 95°C for 5 seconds and annealing/extension at 60°C for 34 seconds.

Relative gene expression levels were calculated using the $2^{-\Delta\Delta Cq}$ method, with GAPDH serving as the internal control. The primer sequences used in the current study were as follows: RIBC2 forward 1, 5'-GACATGAAACCTTTGCTGCTGAA-3' and reverse 1, 5'-CCGAACATCATTATCTGACTGC-3'; RIBC2 forward 2, 5'-AGGCCCTCTACACAGAGACAAG-3' and reverse 2, 5'-GCTTTTTCCTTTCCACTGACTC-3'; GAPDH forward, 5'-ACCCACTCCTCCACCTTTGA-3' and reverse, 5'-CCACCCTGTTGCTGTAGCCA-3'. In the case of RIBC2, the primers forward 1 and reverse 1, which generated a 215 bp product, were used in the initial experiments using 50 paired FFPE tumour and adjacent non-tumour tissues. To further reduce the potential influence of RNA integrity and to assess the robustness of the expression pattern, another RIBC2 primers (forward 2 and reverse 2) yielding a 147 bp product was used, and RT-qPCR was repeated in a subset of 20 paired FFPE tumour and adjacent non-tumour tissues from the same 50-pair cohort.

## Western blot assay

Cells were lysed on ice for 10 minutes using RIPA lysis buffer (Cat. No. P0013, Beyotime) supplemented with 1% protease inhibitor cocktail (Cat. No. ST506, Beyotime). The resulting lysates were centrifuged at 14,000 × g for 5 minutes at 4°C, and the supernatant containing total protein was collected. Protein concentrations were determined using a Bicinchoninic Acid (BCA) Protein Assay Kit (Cat. No. PC0020, Solarbio). Equal amounts of protein were separated on a 10% SDS–polyacrylamide gel (SDS-PAGE) and transferred onto polyvinylidene fluoride (PVDF) membranes. The membranes were blocked with 5% skim milk in TBST (Tris-buffered saline containing 0.1% Tween-20) for 1 hour at room temperature (RT) to prevent nonspecific binding. This was followed by an incubation period at 4°C overnight with diluted primary antibodies, which included RIBC2 Polyclonal antibody (15272–1-AP, 1:1000 dilution, Proteintech), and GAPDH Polyclonal antibody (10494–1-AP, 1:5000 dilution, Proteintech). On the following day, after three washes with TBST (10 minutes each at RT), the membranes were incubated with the corresponding HRP-conjugated secondary antibodies (1:5000 dilution) for 1 hour at RT. The membranes were then washed three additional times with TBST (10 minutes per wash). Protein bands were visualized using an Enhanced Chemiluminescence (ECL) detection kit, and the signal was captured with a gel imaging system (Model C300, Azure Biosystems).

## Cell transfection

The short hairpin RNAs (shRNAs) plasmid targeting RIBC2 and a negative control (NC) shRNA were custom-designed and synthesized by GenCefeBio (Wuxi, China). The target sequences for the shRNAs were as follows: sh-RIBC2–1, GGCTATCAATGACTTCCAACA; sh-RIBC2–2, GCTGCAGTTTGACGAGACAGC; sh-RIBC2–3, GCTGGAGCAGATCCG-CCTAGT. Het-1A-T, TE-10, and OE33 cells were seeded in 6-well plates and allowed to reach approximately 80% confluency within 24 hours. Transfection was carried out using the Lipofectamine™ 3000 Transfection Kit (Cat. No. L3000008, Invitrogen) according to the manufacturer's instructions. Specifically, a transfection mixture containing 2.5 μg of shRNA plasmid, 5 μL of P3000 reagent, 7.5 μL of Lipofectamine™ 3000, and 250 μL of Opti-MEM (Cat. No. 31985−070, Gibco) was gently prepared and incubated at room temperature for 15 minutes before being added dropwise to the cell cultures. After transfection, the cells were incubated at 37°C for 48 hours, after which they were harvested for subsequent experimental analyses.

## Cell doubling time

Het-1A-N and Het-1A-T cells were seeded separately into 12-well plates at a density of $1 \times 10^4$ cells per well and cultured in standard growth medium. Cell proliferation was monitored over a 7-day period, with cell counts from three parallel wells recorded daily using a CytoSMART™ Cell Counter (Corning, USA). The population doubling time (TD) was calculated using the following formula: $TD = t \times \log2/(\log N_t - \log N_i)$; where t is duration, $N_t$ is cell number on Day t, $N_i$ is initial cell number.

## Soft agar colony formation assay

A base layer was first prepared by mixing 1.2% sterile low–melting point agarose with 2×DMEM supplemented with FBS in a 1:1 ratio, and 1 mL of this mixture was added to each well of a 6-well plate. For the upper layer, cells were suspended in a mixture of 0.7% low–melting point agarose and 2×DMEM containing FBS, which was then gently overlaid onto the solidified base layer. After 14 days of incubation, cell colonies were visualized and documented using a light microscope (TS2, Nikon, Japan).

## Cell counting kit-8 (CCK-8) assay

Transfected Het-1A-T, TE-10, and OE33 cells (as described above) were seeded into 96-well plates, and cell viability was evaluated using the CCK-8 assay (Cat. No. MF128−01, Mer5bio) following the manufacturer's protocol. At designated time points, CCK-8 reagent was added to each well and incubated at 37°C for 2 hours. The absorbance was then measured at 450 nm using a microplate reader (Multiskan FC, Thermo Fisher Scientific, USA) to determine cell viability.

## Colony formation assay

Het-1A-N and Het-1A-T cells, as well as transfected Het-1A-T, TE-10, and OE33 cells (as described previously), were seeded into 6-well plates and cultured for 7–14 days to allow colony formation. After incubation, the cells were fixed with methanol for 15 minutes and subsequently stained with 0.1% (v/v) crystal violet (Cat. No. G1063, Solarbio) for another 15 minutes at RT. Colonies containing more than 50 cells were manually counted, and the average colony number was used to quantify the colony-forming ability of the cells, reflecting their proliferative potential.

## Wound healing assay

Het-1A-N and Het-1A-T cells, as well as transfected Het-1A-T, TE-10, and OE33 cell lines (as described previously), were seeded into 6-well plates to form a confluent monolayer. A uniform scratch was then created using a 200 µL pipette tip to generate a wound gap in each well. The wells were subsequently washed three times with 1×PBS to remove detached cells and incubated in serum-free medium at 37°C. Cell migration into the wound area was observed and photographed under a light microscope (Model TS2, Nikon, Japan) at 0 and 24 hours. The extent of wound closure was quantified using ImageJ software (1.51a, National Institutes of Health).

## Transwell invasion assay

A Transwell chamber system (Cat. No. 3422, Corning) was employed to evaluate the invasive capacity of cells. Briefly, Matrigel (Cat. No. 354234, Corning) was thawed at 4°C, and 50 µL of a Matrigel–medium mixture (ratio 1:8) was added to pre-chilled Transwell inserts and allowed to solidify at 37°C for 1 hour. Subsequently, Het-1A-N and Het-1A-T cells, or transfected Het-1A-T, TE-10, and OE33 cells (as described earlier), were seeded into the upper chamber containing serum-free Opti-MEM (Cat. No. 31985−070, Gibco). The lower chamber was filled with complete growth medium supplemented with 20% FBS, 100 U/mL penicillin, and 0.1 mg/mL streptomycin to serve as a chemoattractant. After 24 hours of incubation at 37°C in a 5% $CO_2$ atmosphere, non-invading cells remaining on the upper surface of the membrane were gently removed with a cotton swab. The inserts were then fixed with methanol for 15 minutes and stained with 1% (v/v) crystal violet (Cat. No. G1062, Solarbio) for an additional 15 minutes at RT. Representative fields were imaged under a light microscope (Model DM4B, Leica, Germany), and the number of invaded cells per field was manually counted to quantify cell invasion.

## In vivo xenograft assay

Four-week-old female BALB/c nude mice were obtained from HuafukangBio (Beijing, China) and housed under specific pathogen-free (SPF) conditions. Het-1A-N, Het-1A-T, KYSE-450 and OE33 cells were cultured, resuspended in sterile

PBS, and subcutaneously injected into the mice at a concentration of $1 \times 10^7$ cells per mouse. Tumor growth was monitored, and tumor volume was measured on day 28 post-injection, calculated using the formula: volume = length×(width$^2$)/2. At the end of the experiment, mice were euthanized by cervical dislocation under deep anesthesia induced with sodium pentobarbital (40 mg·kg$^{-1}$, i.p.). Anesthetic depth was verified by the absence of pedal and palpebral reflexes. Death was confirmed by the absence of heartbeat and corneal reflex for ≥2 min. Tumors were then excised and weighed for further analysis. All animal experiments were carried out with approved protocols, relevant guidelines and regulations of the Laboratory Animal Ethics Committee of North China University of Technology (protocol code: 2023-SY-070). All methods were reported in accordance with ARRIVE guidelines.

### H&E staining

Het-1A-N and Het-1A-T cells were fixed in 95% ethanol for 10 minutes. For tissue sections, samples were first incubated at 60°C for 1 hour, followed by dewaxing in xylene and rehydration through a graded series of ethanol solutions. Subsequently, both cells and tissue sections were stained with hematoxylin for 3 minutes, subjected to brief differentiation in 1% hydrochloric acid alcohol, and then counterstained with eosin for 2 minutes. After washing with PBS, the samples were mounted using neutral gum. All staining procedures were performed at room temperature (RT). After staining, the cells and tissue sections were examined and imaged using a light microscope (DM4B, Leica, Germany).

### Immunohistochemistry (IHC)

Paraffin-embedded tissue sections were first incubated at 60°C for 1 hour to melt excess paraffin, followed by deparaffinization in xylene I and II for 10 minutes each. The sections were then rehydrated through a graded series of ethanol solutions (100%, 90%, 80%, and 70%, each for 5 minutes) and rinsed three times with PBS at RT for 5 minutes per wash. To quench endogenous peroxidase activity, the sections were treated with 3% hydrogen peroxide ($H_2O_2$) for 8 minutes, followed by antigen retrieval using a sodium citrate buffer. After retrieval, nonspecific binding was blocked with 5% bovine serum albumin (BSA) (Cat. No. AR0004, Bosterbio) at 37°C for 20 minutes. Subsequently, the tissue sections were subjected to overnight incubation at 4°C with the following primary antibodies: KI67 Polyclonal antibody (27309–1-AP, 1:10000, Proteintech) and pan-keratin Polyclonal antibody (26411–1-AP, 1:4000, Proteintech). After washing, sections were incubated with HRP-conjugated secondary antibodies (Cat. No. PV-6001, ZSGBbio) for 40 minutes at RT, followed by color development using 3,3'-diaminobenzidine (DAB) and counterstaining with hematoxylin. Finally, the slides were mounted with neutral gum, and the stained sections were examined and imaged under a light microscope (DM4B, Leica, Germany).

### Statistical analysis

All statistical analyses were conducted using R software (version 4.2.2). Comparisons between two groups were performed using either the Wilcoxon rank-sum test or the Student's t-test, depending on data distribution. The chi-squared test was applied to assess differences in categorical clinical variables. Kaplan–Meier survival curves and ROC analyses were generated to evaluate the prognostic and diagnostic significance of candidate biomarkers. All biological experiments were carried out independently in triplicate, and a $p$-value $< 0.05$ was considered statistically significant unless otherwise indicated.

## Results

### Establishment of a chronic in vitro model of esophageal epithelial malignant transformation

To mimic the malignant transformation of esophageal epithelial cells, an in vitro model was established by exposing non-tumorigenic Het-1A cells to chronic low-dose carcinogenic stimulation. After 50 consecutive passages, the

resulting transformed cells (Het-1A-T) displayed distinct morphological and phenotypic changes compared with the non-transformed controls (Het-1A-N). Specifically, Het-1A-T cells exhibited enlarged and irregular morphology, enhanced intercellular adhesion, cytoplasmic vacuolization, and the appearance of multinucleated giant cells compared to Het-1A-N cells (Fig 1A). Cell proliferation analysis revealed that Het-1A-T cells had a significantly higher growth rate (Fig 1B) and a shorter doubling time (Fig 1C) than Het-1A-N cells. Functional assays including soft agar colony formation, plate colony formation, wound healing, and Transwell invasion further confirmed that Het-1A-T cells possessed markedly enhanced migratory and invasive abilities compared to their non-transformed counterparts (Fig 1D–1G). In in vivo tumorigenicity assays using nude mice, Het-1A-T cells, similar to the positive control ESCC (KYSE-450) and EAC (OE33) cells formed subcutaneous tumors, while Het-1A-N cells failed to do so (Fig 1H–1I). Het-1A-T cells generated tumours with malignant histology and high Ki-67 and pan-CK expression, confirming that transformed squamous epithelial cells acquired tumourigenic capacity in vivo (Fig 1H–K). Collectively, these findings confirm the successful establishment of a malignant transformation model, providing a robust system for subsequent mechanistic and transcriptomic analyses.

## Identification of candidate diagnostic biomarkers for early EC

To identify potential biomarkers for the early detection of EC, we first examined transcriptomic alterations associated with malignant transformation in vitro. A total of 1,580 DEGs were identified between Het-1A-T and Het-1A-N cells, comprising 791 upregulated and 789 downregulated genes (S1A and S1B Fig). To enhance the clinical relevance of our findings, transcriptomic data from the TCGA-EC cohort were integrated, yielding 3,510 DEGs between tumor and normal esophageal tissues (S1C and S1D Fig). Cross-comparison of the in vitro and in vivo datasets revealed 481 overlapping DEGs (S1E Fig). Univariate Cox regression analysis of these overlapping genes identified 22 DEGs significantly associated with patient prognosis, including *HSPA1B, STK32A, GLTPD2, ABCC6, ESM1, COX6B2, APLN, BTG2, CACNG8, DUSP2, RIPPLY3, SLIT2, SLIT3, RIBC2, CXCL8, PRR18, HSPA1A, NOL4, SLC24A4, PHYHIP, ACP7*, and *LMF1* (S2 Table). Functional enrichment analysis indicated that these genes were predominantly involved in signal transduction, cell proliferation, and cytokine-mediated signaling pathways (S1F Fig). Using these 22 prognostic DEGs, we constructed and evaluated four machine learning models for EC diagnosis. Among them, the logistic regression model exhibited the best performance, achieving an AUC of 1.0 in the TCGA training cohort and 0.84 in the GEO validation cohort (Fig 2A–2D). Feature importance analysis revealed that *ESM1, ABCC6, CXCL8, RIBC2*, and *APLN* were the top five contributors to the model (Fig 2E and 2F). Further SHAP dependency analysis demonstrated that elevated expression of *ESM1, CXCL8, RIBC2*, and *APLN* corresponded to higher SHAP values, indicating a positive association with EC diagnosis, while *ABCC6* expression showed a negative correlation, suggesting a protective effect (Fig 2G). Overall, these findings identify *ESM1, ABCC6, CXCL8, RIBC2*, and *APLN* as robust diagnostic biomarkers with strong potential for early-stage EC detection.

## Identification of candidate prognostic biomarkers for EC

To identify robust prognostic biomarkers for EC, the initial set of 22 prognosis-related DEGs was further refined using LASSO regression analysis, resulting in the identification of 17 key prognostic genes: *HSPA1B, STK32A, ABCC6, ESM1, COX6B2, APLN, BTG2, CACNG8, RIPPLY3, SLIT2, SLIT3, RIBC2, CXCL8, HSPA1A, SLC24A4, ACP7*, and *LMF1* (Fig 3A). A risk score was computed for each EC patient based on the expression levels and LASSO-derived coefficients of these 17 genes, and patients were subsequently categorized into high-risk (risk score^high) and low-risk (risk score^low) groups (Fig 3B). In the TCGA-EC cohort, Kaplan–Meier survival analysis revealed that patients in the low-risk group had significantly longer overall survival compared to those in the high-risk group (Fig 3C). Given the different biological backgrounds of ESCC and EAC, we further performed Kaplan-Meier analyses stratified by histological subtype, and it was also found that low-risk group had better survival than high-risk group in both TCGA-ESCC and TCGA-EAC cohorts (S2 Fig). The predictive accuracy of this prognostic model was further validated using time-dependent ROC curves, yielding AUC values of 0.83, 0.84, and 0.79 for 1-, 2-, and 3-year survival, respectively (Fig 3D). Notably, patients with advanced

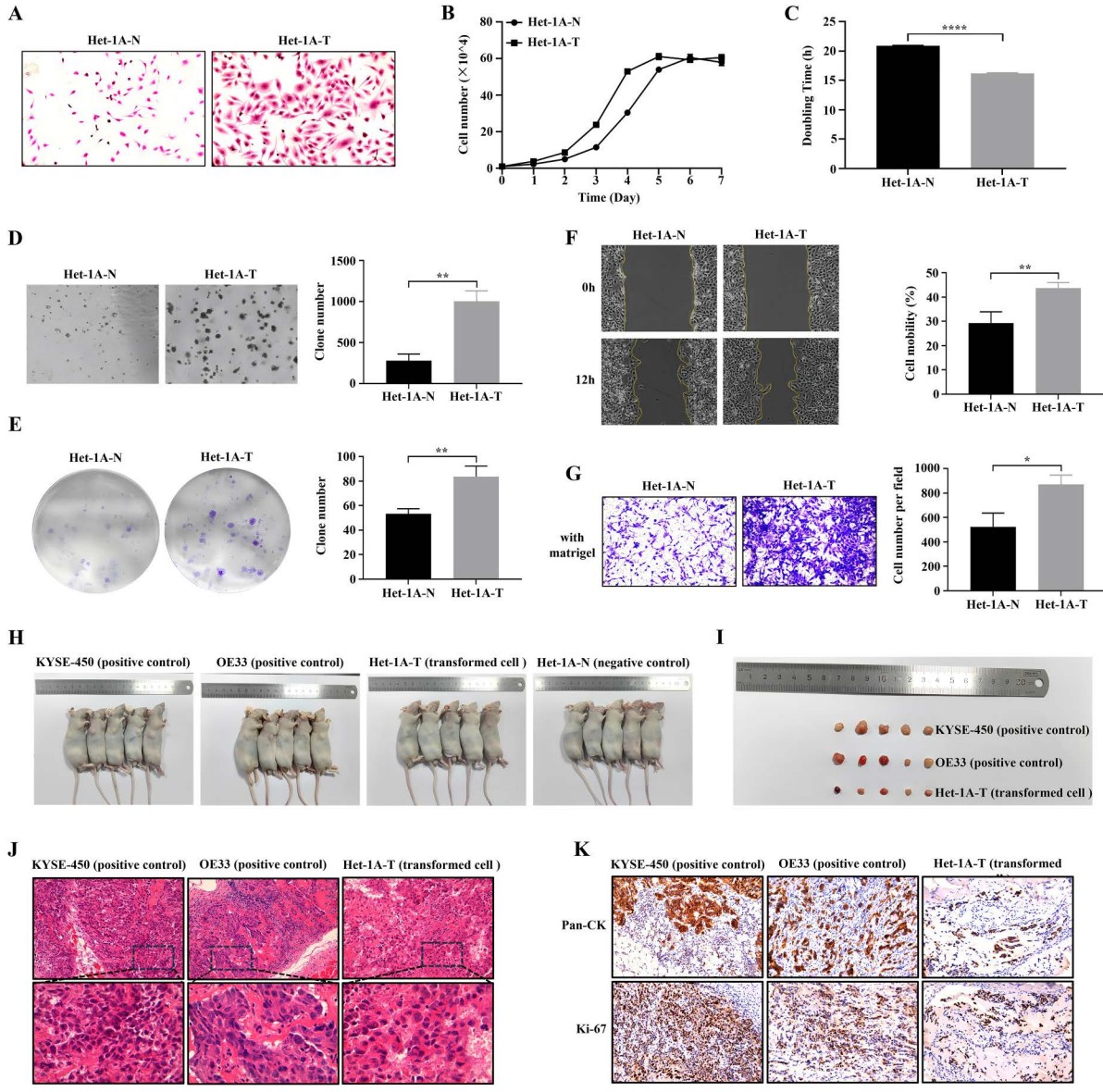

**Fig 1. Establishment of an in vitro model of esophageal epithelial malignant transformation.** (A) Representative H&E-stained images showing distinct morphological differences between non-transformed Het-1A-N and transformed Het-1A-T cells (magnification, ×100). (B, C) Cell growth curves (B) and doubling time (C) of Het-1A-N and Het-1A-T cells. (D-G) Evaluation of malignant characteristics in Het-1A-N and Het-1A-T cells using soft agar colony formation (D, ×40), colony formation (E), wound healing (F, ×40), and Transwell invasion (G, ×100) assays. (H) Images of nude mice bearing subcutaneous xenografts generated from KYSE-450 (ESCC positive control), OE33 (EAC positive control), Het-1A-T (transformed squamous epithelial cells) and Het-1A-N (negative control). (I) Photographs of resected xenograft tumours from KYSE-450, OE33 and Het-1A-T groups; no palpable tumours were formed by Het-1A-N cells. (J) Representative H&E staining of xenograft tumours derived from KYSE-450, OE33 and Het-1A-T cells; higher-magnification views of the boxed areas are shown in the lower panels. (K) Representative immunohistochemical staining of xenograft tumors showing Pan-CK and Ki-67 expression (magnification, ×200). *P < 0.05, **P < 0.01 and ****P < 0.0001.

nodal (N1–N3) or distant metastatic (M1) disease exhibited significantly higher risk scores than those with N0 or M0 status (Fig 3E and 3F), suggesting that the prognostic gene signature is reflective of tumor aggressiveness and metastatic potential. Further immune profiling revealed distinct tumor microenvironment characteristics between the high- and

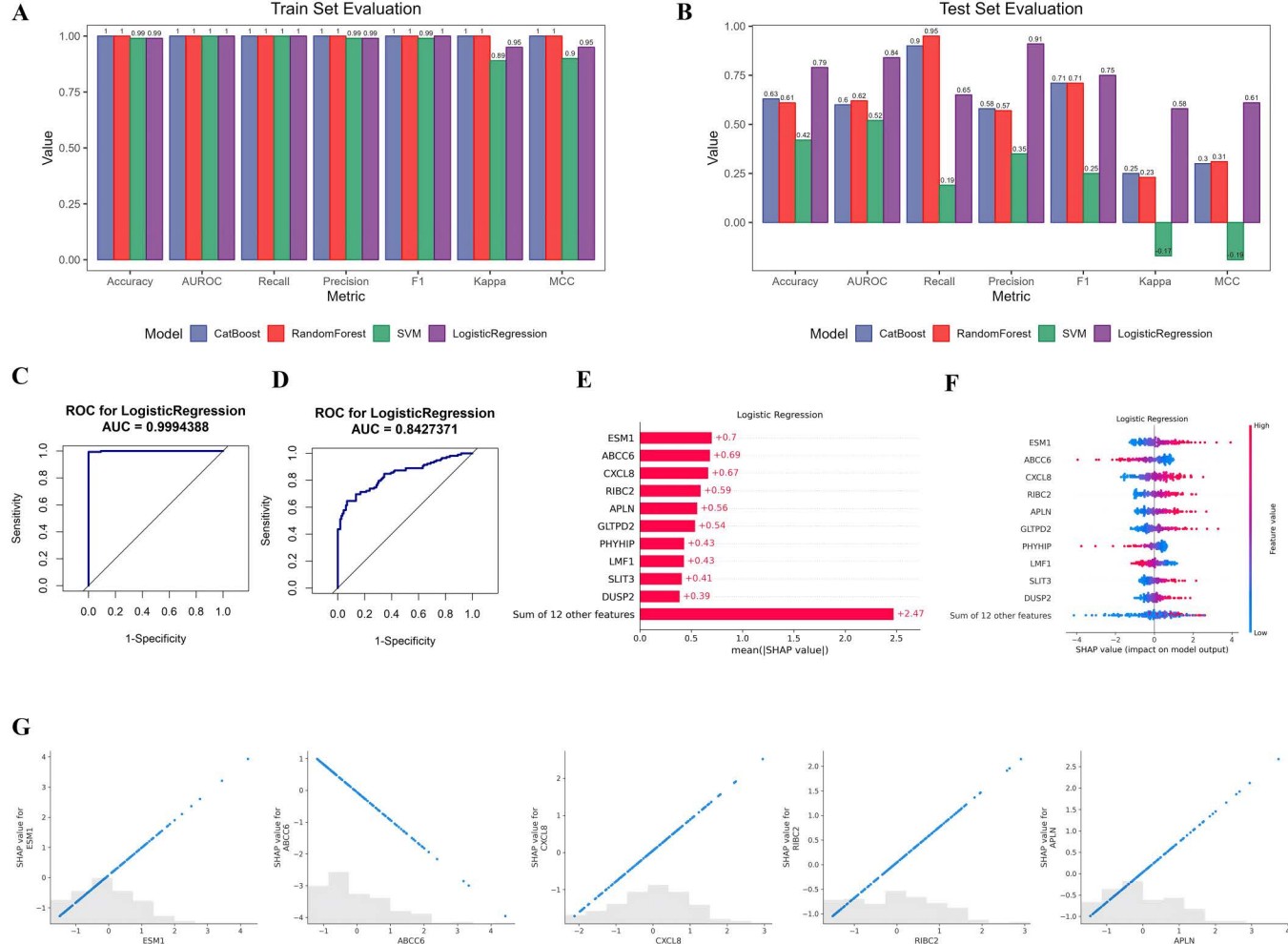

**Fig 2. Construction and interpretation of a diagnostic model for EC based on machine learning.** (A, B) Comparative performance assessment of four machine learning algorithms (CatBoost, Random Forest, Support Vector Machine and Logistic Regression) in the training (A) and testing (B) cohorts. (C, D) ROC curves and corresponding AUC values for the logistic regression model in the training (C) and testing (D) datasets. (E) Mean SHAP value plot showing the top 10 most influential features contributing to the logistic regression model. (F) SHAP summary plot illustrating the distribution and magnitude of SHAP values for all features, ranked according to their impact on model predictions. (G) SHAP dependence plots for the top five diagnostic genes, demonstrating the relationship between gene expression levels and their influence on EC classification. Positive SHAP values represent a stronger contribution toward EC prediction.

low-risk groups, underscoring the immunological relevance of this prognostic model. The low-risk group (risk score^low) displayed significant enrichment of immune-related functions, including type I/II interferon (IFN) responses, immune checkpoint activity, para-inflammation, and increased HLA gene expression (Fig 3G). In addition, the infiltration of both innate and adaptive immune cells such as immature dendritic cells (iDCs), macrophages, T helper cells, and natural killer (NK) cells was markedly higher in the low-risk group (Fig 3H). To elucidate the molecular mechanisms underlying the distinct phenotypes between risk groups, GSEA was performed. The high-risk group (risk score^high) was characterized by the enrichment of metabolic pathways, including lipid absorption and sterol homeostasis (Fig 3I), whereas the low-risk group showed predominant enrichment of immune-related pathways, such as immune response-regulating signaling pathways, leukocyte-mediated immunity, and the T cell receptor complex (Fig 3J). Together, these results suggest

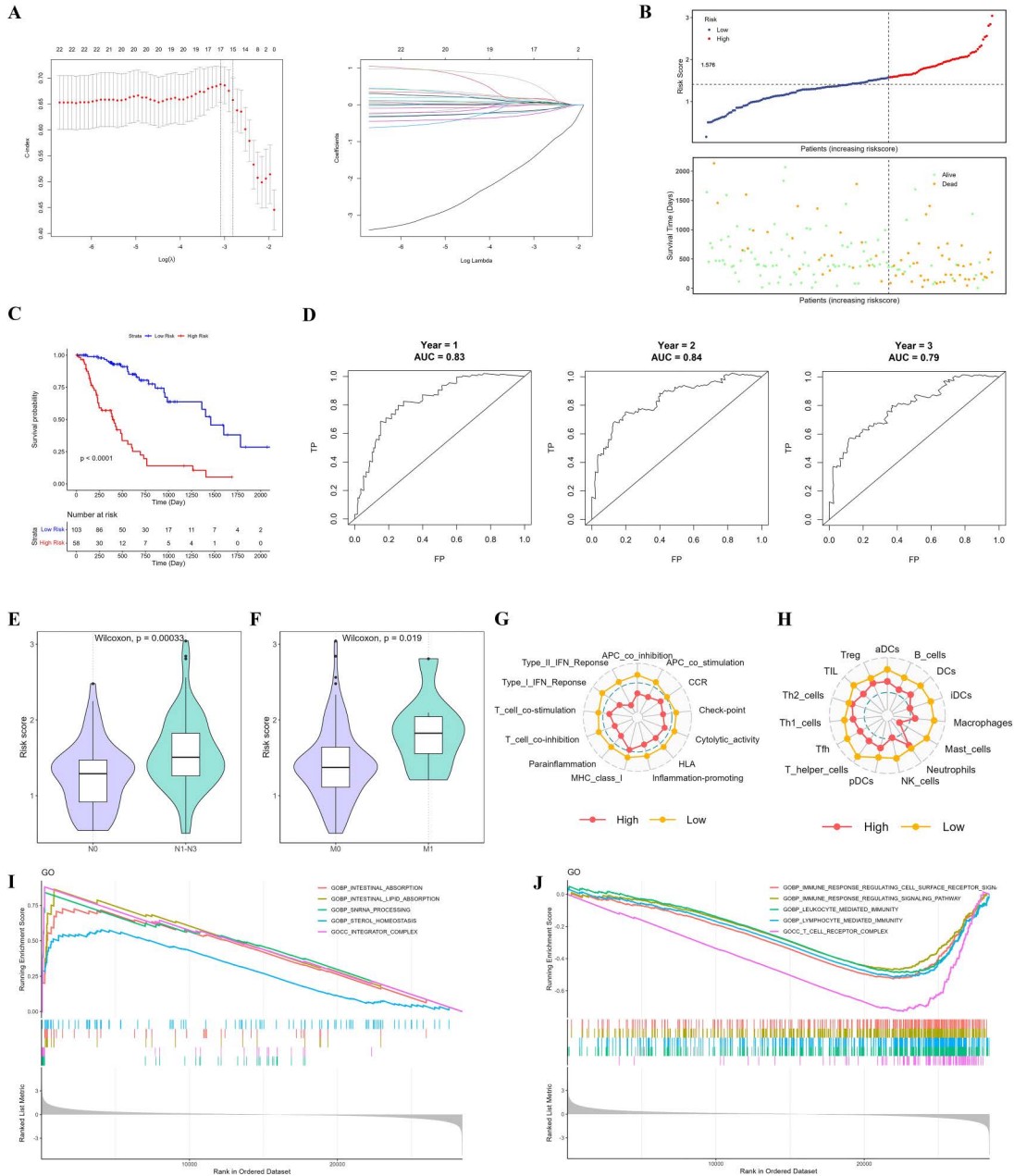

**Fig 3. Construction and evaluation of a prognostic risk model for EC.** (A) LASSO Cox regression analysis of 22 prognosis-associated DEGs, with the optimal lambda value determined through 10-fold cross-validation. (B) Distribution of survival status among patients in the low- and high-risk groups. (C) Kaplan–Meier survival analysis indicating significantly improved overall survival in the low-risk group compared to the high-risk group. (D) Time-dependent ROC curves demonstrating the predictive accuracy of the risk model for 1-, 2-, and 3-year overall survival. (E, F) Box plots comparing risk scores across N (E) and M (F) staging categories. (G, H) Radar plots illustrating differences in immune function enrichment (G) and immune cell infiltration (H) between the low- and high-risk groups. (I, J) GSEA showing significant enrichment of metabolism-related pathways in the high-risk group (I) and immune-related pathways in the low-risk group (J).

a strong association between metabolic reprogramming, immune exclusion, and poor prognosis in EC, indicating that disrupted immune regulation and altered metabolism may jointly drive disease progression.

## Identification of *RIBC2* as a potential diagnostic and prognostic biomarker

Among the diagnostic and prognostic biomarkers identified, *RIBC2* expression was found to be significantly upregulated in both transformed Het-1A-T cells (Fig 4A) and esophageal tumor tissues (Fig 4B). Kaplan–Meier survival analysis demonstrated that high *RIBC2* expression was strongly associated with poorer overall survival in EC patients (Fig 4C). Additionally, it was observed that the adverse prognostic impact of RIBC2 high expression was significant in EAC, whereas in ESCC the Kaplan-Meier curves did not show a statistically significant difference between the low- and high-RIBC2 expression groups (S3 Fig). Furthermore, ROC curve analysis confirmed the diagnostic utility of *RIBC2* in EC, yielding an AUC of 0.793 (Fig 4D). Specifically, RIBC2 showed diagnostic potential in EC with AUCs of 0.772 for EAC and 0.812 for ESCC, respectively (S4 Fig). Immune correlation analysis revealed that patients with elevated *RIBC2* expression exhibited higher levels of immune checkpoint molecules such as *CD276* and *NT5E*, while showing reduced expression of *IL6R* (Fig 4E). In the context of the cancer–immunity cycle, high *RIBC2* expression was linked to enhanced activity in the recognition of cancer cells by T cells phase, but decreased activity in the immune cell infiltration into tumors phase (Fig 4F). Additionally, these patients demonstrated significantly higher TIDE exclusion scores, indicative of an immune-excluded phenotype (Fig

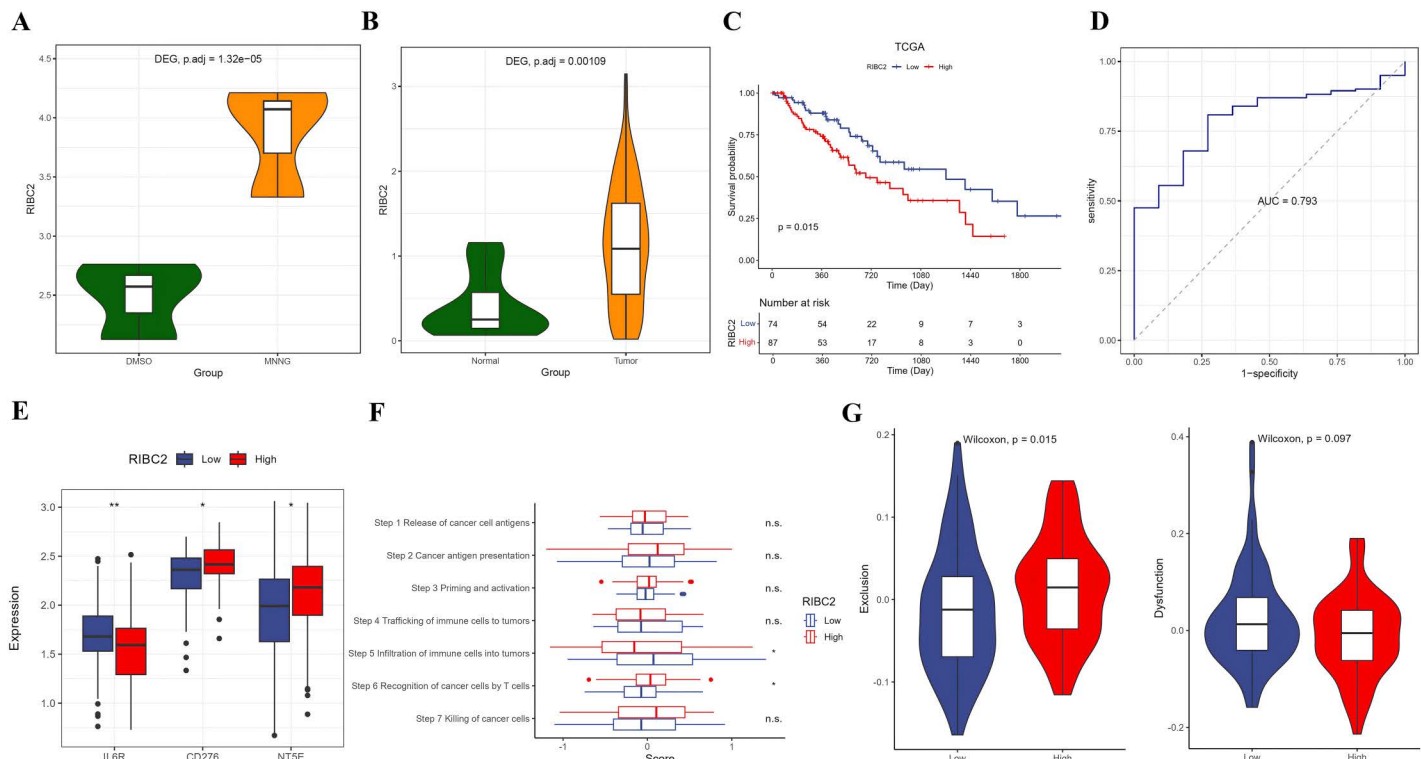

**Fig 4. RIBC2 in diagnosis, prognosis and immunity of EC.** (A, B) Differential expression of RIBC2 between Het-1A-T and Het-1A-N cells (A) and between tumor and normal tissues in the TCGA-EC cohort (B). (C) Kaplan-Meier survival curves demonstrating that high RIBC2 expression correlates with poorer overall survival in EC patients. (D) ROC curve illustrating the diagnostic accuracy of RIBC2 expression for distinguishing EC from normal tissues. (E) Comparison of immune checkpoint gene expression between high- and low-RIBC2 expression groups. (F) Correlation analysis between RIBC2 expression and the cancer–immunity cycle. (G) Comparison of TIDE exclusion and dysfunction scores across high- and low-RIBC2 expression groups, reflecting differences in potential immunotherapy responsiveness.

). We also found that the infiltration of CD4 Tem and CD8 T cells were significantly higher in the low RIBC2 expression group, while the abundance of Th2 cells were much higher in the high RIBC2 expression group (S5A Fig). Further GSEA analysis showed that pyrimidine metabolism, cell cycle, mismatch repair and DNA replication were activated, while O-glycan biosynthesis and sphingolipid metabolism were suppressed in the high RIBC2 expression group (S5B Fig). More-over, we identified a total of 1,255 DEGs between low- and high-RIBC2 expression groups (S5C Fig). These DEGs were significantly enriched into KEGG pathways of neuroactive ligand-receptor interaction, cornified envelope formation, nicotine addition, biological processes of immune response and peptidase activity (S5D Fig). Moreover, by PPI analysis, we found that RIBC2 may interact with TEKT2, C5orf49 and CFAP299 (S5E Fig). Collectively, these findings suggest that *RIBC2* serves as both a diagnostic and prognostic biomarker for EC and may also play a crucial role in modulating the tumor immune microenvironment, potentially contributing to immune evasion and reduced responsiveness to immunotherapy.

We subsequently validated the expression pattern and clinical relevance of *RIBC2* using both in vitro models and clinical samples. As shown in Fig 5A and 5B, *RIBC2* expression was significantly higher in Het-1A-T cells compared with Het-1A-N cells. Similarly, elevated *RIBC2* levels were observed in multiple EC cell lines including KYSE-410, TE-10, and

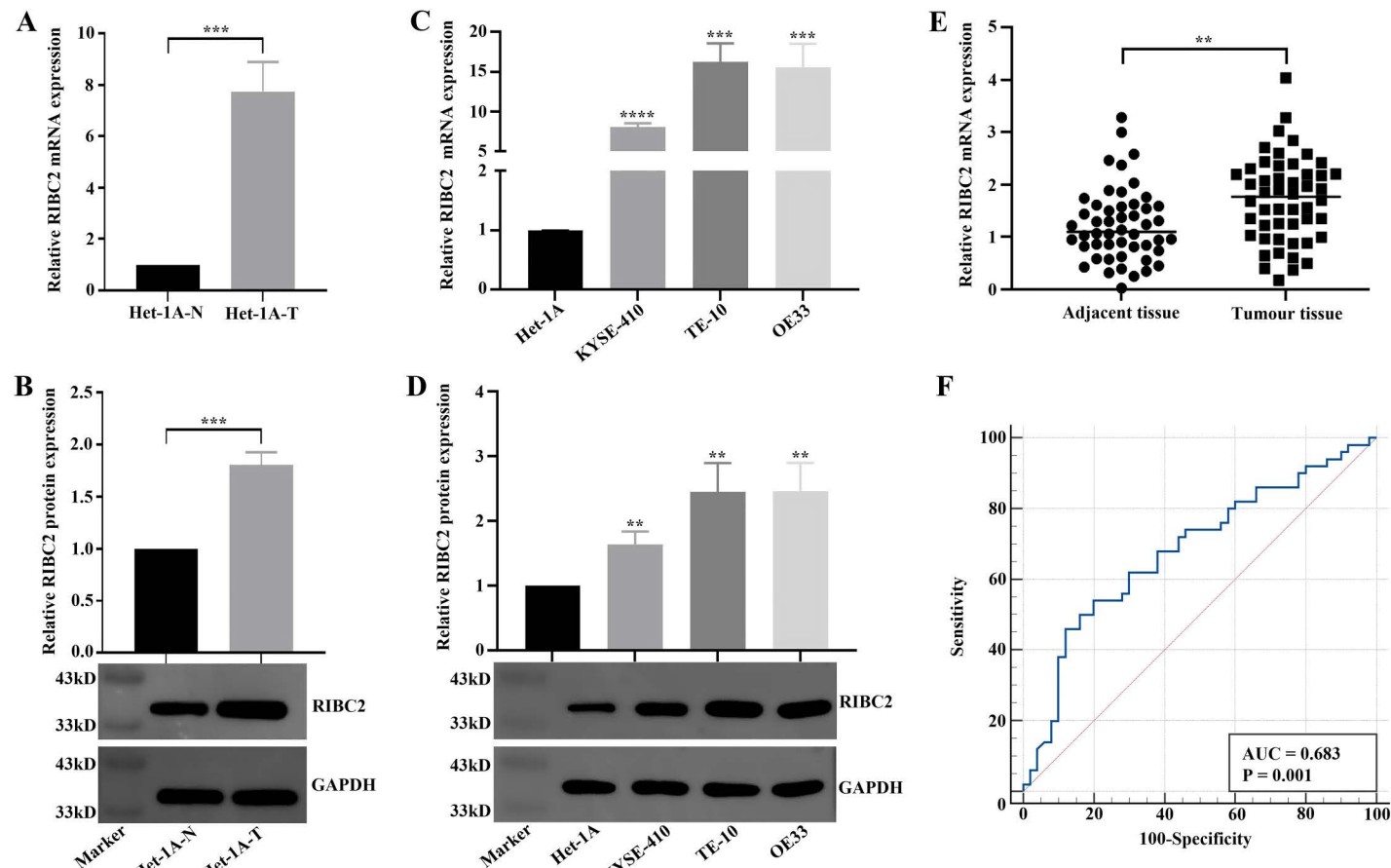

**Fig 5. Validation of RIBC2 expression and its clinical relevance in EC.** (A, B) Quantification of RIBC2 mRNA (A) and protein (B) expression levels in Het-1A-N and Het-1A-T cells. (C, D) Comparison of RIBC2 mRNA (C) and protein (D) expression levels between the normal esophageal epithelial cell line (Het-1A) and EC cell lines (KYSE-410, TE-10, and OE33). (E) Analysis of RIBC2 mRNA expression in 50 paired EC tissues and their corre-sponding adjacent normal tissues. of EC. (F) ROC curve illustrating the diagnostic performance of tissue RIBC2 expression. **P < 0.01, ***P < 0.001 and ****P < 0.0001.

OE33 relative to the non-tumorigenic esophageal epithelial cell line Het-1A (Fig 5C and 5D). To validate these findings in vivo, RT-qPCR analysis was conducted on paired tumor and adjacent non-tumor tissues from EC patients. Consistent with the cell line data, *RIBC2* expression was markedly upregulated in tumor tissues (Figs 5E and S6). Clinicopathological correlation analysis revealed that high *RIBC2* expression was significantly associated with older patient age and advanced tumor stage (S3 Table). Furthermore, ROC curve analysis demonstrated that *RIBC2* effectively distinguished EC tissues from adjacent normal tissues, yielding an AUC of 0.683 (Fig 5F). Collectively, these findings confirm that *RIBC2* is consistently overexpressed in EC and is correlated with unfavorable clinical characteristics, highlighting its potential as a clinically relevant biomarker.

## Promotion of malignant phenotypes in Het-1A-T cells by RIBC2

Since *RIBC2* was found to be significantly upregulated in transformed cells, we next examined its functional role in malignant Het-1A-T cells, using TE-10 and OE33 EC cell lines as positive controls. Three short hairpin RNAs (shRNAs) targeting *RIBC2*, sh-*RIBC2*–1, sh-*RIBC2*–2, and sh-*RIBC2*–3 were designed, all of which effectively reduced *RIBC2* mRNA expression (Fig 6A). Among them, sh-*RIBC2*–2 (hereafter referred to as sh-*RIBC2*) exhibited the most efficient knockdown and was selected for further analysis. Western blotting confirmed a marked decrease in RIBC2 protein levels following transfection (Fig 6B). Functionally, *RIBC2* silencing significantly suppressed cell proliferation, as demonstrated by CCK-8 and colony formation assays (Fig 6C and 6D). Moreover, wound healing and Transwell invasion assays revealed that RIBC2 knockdown notably reduced the migratory and invasive abilities of Het-1A-T, TE-10, and OE33 cells (Fig 6E and 6F). Taken together, these results demonstrate that *RIBC2* promotes the proliferative, migratory, and invasive behavior of transformed esophageal epithelial cells, underscoring its potential role as a key driver of malignant transformation in EC.

## Discussion

EC continues to be a malignancy with a poor prognosis, primarily due to the lack of reliable early diagnostic approaches and an incomplete understanding of the molecular mechanisms underlying tumor initiation. In this study, we utilized a well-established in vitro malignant transformation model of esophageal epithelial cells to examine the molecular processes involved in EC tumorigenesis, a system that has been extensively applied in both in vitro and in vivo research [17–19]. The transformed cells demonstrated markedly increased proliferation, migration, invasion, and tumor-forming ability in nude mice, consistent with previous findings. This model thus provides a robust experimental platform for transcriptomic profiling and the identification of key molecular drivers contributing to EC development.

Transcriptome sequencing of normal and transformed esophageal epithelial cells, combined with publicly available transcriptomic datasets from EC tissues, facilitated the identification of key candidate biomarkers using a machine learning–based diagnostic framework. Among the identified genes, the top five contributors to diagnostic accuracy were *ESM1*, *ABCC6*, *CXCL8*, *RIBC2*, and *APLN*. Notably, *ESM1* has been reported to be highly expressed across various cancer types including ovarian, gastric, breast, esophageal, and cervical cancers and is known to promote tumor growth and progression primarily through its roles in angiogenesis and fatty acid metabolism [20–24]. In EC, *ESM1* expression is markedly upregulated in tumor tissues and shows a positive correlation with advanced tumor stage and poorer survival outcomes. Functional studies have demonstrated that *ESM1* knockdown can suppress EC cell proliferation, migration, and invasion [23]. *ABCC6,* a member of the ATP-binding cassette (ABC) transporter family, plays diverse roles in human diseases. For instance, *ABCC6* deficiency leads to pseudoxanthoma elasticum, a disorder characterized by ectopic calcification [25]. In hepatocellular carcinoma, *ABCC6* silencing has been shown to enhance cell proliferation and suppress apoptosis through activation of the PPAR signaling pathway [26]. Moreover, *ABCC6* has been implicated in the regulation of lipid and cholesterol metabolism [27,28]. However, its specific function in EC remains poorly understood. A recent study by Zhang et al. suggested that *ABCC6* may serve as a potential target of cranberry proanthocyanidins, which exhibit inhibitory effects on EC growth [29]. The significance of *CXCL8* (also known as interleukin-8) has been well established in

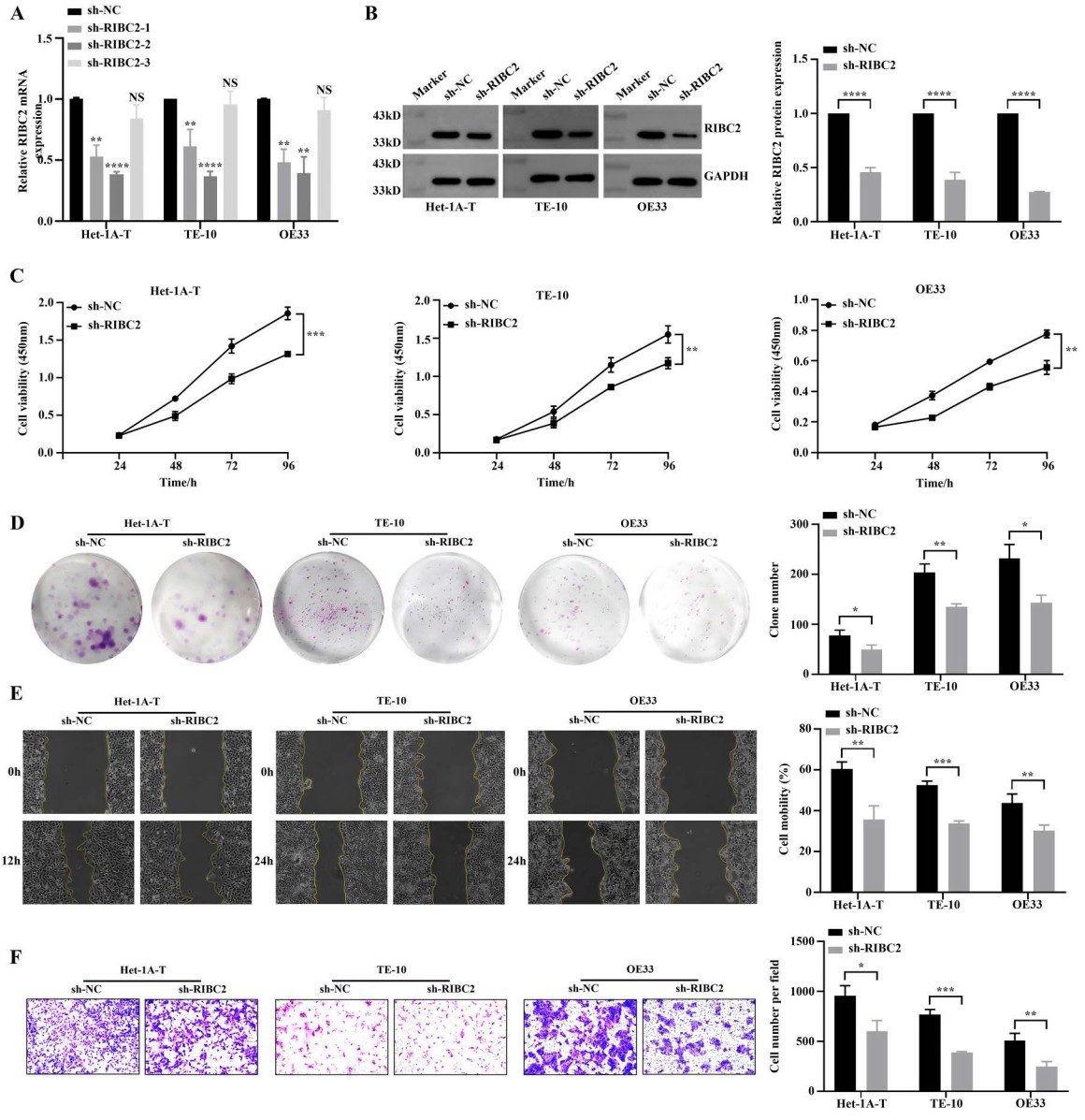

**Fig 6. RIBC2 promotes malignant transformation of human esophageal epithelial cells.** (A, B) Quantification of RIBC2 mRNA (A) and protein (B) expression levels following transfection with RIBC2-specific shRNA or negative control shRNA. (C-F) Functional assessment of cell proliferation (C), colony formation (D), migration (magnification, 40×) (E), and invasion (magnification, 100×) (F) in RIBC2 knockdown and control groups. $^{NS}$P > 0.05, *P < 0.05, **P < 0.01, ***P < 0.001 and ****P < 0.0001.

both inflammatory disorders and cancer biology. It is a cytokine secreted by immune, stromal, and tumor cells, playing a pivotal role in tumor initiation, progression, and therapeutic response through interactions with its specific receptors [30]. The molecular functions of *CXCL8* in cancer development and therapy have been extensively characterized. *CXCL8* promotes cell proliferation, invasion, and survival, while also contributing to tumor stemness, angiogenesis, and remodeling of the tumor microenvironment through activation of the PI3K/Akt, MAPK, JAK/STAT3, and FAK/Src signaling pathways, as comprehensively reviewed by Xiong et al [31]. APLN binds to its receptor APLNR and participates in the pathogenesis of

cancers [32], heart diseases [33], reproduction dysfunction [34] and diabetes [35]. In EC, *APLN* has been shown to influence cell proliferation, apoptosis, migration, and stemness, with several studies exploring the underlying signaling mechanisms [36,37]. Regarding *RIBC2*, it has been identified as prognostic markers in liver and cervical cancers by bioinformatic analysis [38,39]. Higher expressions of RIBC2 were associated with better survival in cervical cancer, but worse outcome in liver cancer. However, its biological function and molecular mechanisms in cancers remain unclear. It has been reported that RIBC2 is expressed in motile cilia and plays a crucial role in ciliary movement by interacting with YbX2 [40], which could modulate mRNA stability, stemness and chemoresistance in endometrial cancer [41,42].

However, to date, the biological role and mechanistic significance of RIBC2 in EC remain largely unexplored, representing an important gap in current research. Accordingly, we conducted bioinformatic, clinical, and in vitro investigations to elucidate the role of *RIBC2* in EC. Transcriptomic analyses revealed that *RIBC2* expression was markedly elevated in both EC tissues and MNNG-induced malignantly transformed esophageal epithelial cells, with its expression levels closely correlated with EC onset and progression. These observations were subsequently validated using clinical specimens, confirming the upregulation and clinical relevance of *RIBC2* in EC. Functional experiments further demonstrated that *RIBC2* modulates key malignant phenotypes, including cell proliferation, migration, and invasion, in transformed esophageal cells and in both histological subtypes (ESCC and EAC cell lines). These results suggest that MNNG may contribute to esophageal carcinogenesis at least in part via *RIBC2* upregulation. We note that ESCC and EAC differ in cell of origin, risk factors and molecular architectures. Although RIBC2 knockdown produced similar phenotypes across ESCC (KYSE-410, TE-10) and EAC (OE33) cells, the underlying pathways may differ by histology. Future work using subtype-matched 3D organoids or in vivo models, together with patient-level analyses stratified by histology will be important to define subtype-specific mechanisms and clinical relevance of *RIBC2*. In addition, to validate the molecular mechanisms of RIBC2, we will confirm the putative interaction network by co-immunoprecipitation (co-IP) for candidate partners (TEKT2, C5orf49, CFAP299) in ESCC (KYSE-410, TE-10) and EAC (OE33) cells, followed by subcellular localization. Second, guided by our enrichment results, we will determine pathway-related proteins (e.g., keratinization/cornified-envelope markers such as KRT5/KRT14; neuroactive ligand–receptor nodes including nAChR subunits with downstream p-ERK/p-AKT) to evaluate whether pathway modulation attenuates RIBC2-dependent proliferation/migration/invasion. Furthermore, patients with low versus high RIBC2 expression exhibit differences in the immune microenvironment and in activated/ suppressed pathways, suggesting that therapeutic strategies may need to be tailored by RIBC2 status. Multi-omics, including genomics, epigenomics, transcriptomics, proteomics/ phospho-proteomics, metabolomics, and radiomics, provides a useful tool to translate laboratory findings into therapeutically relevant biomarkers [43,44]. Briefly, integrating multi-omics with network pharmacology can identify druggable pathways and network nodes associated with RIBC2, and develop subtype-specific strategies for patients with high versus low RIBC2 expression. Thus, in the future work, we will stratify ESCC and EAC cohorts by RIBC2 expression and perform multi-omics profiling to refine pathway signatures in the high- versus low-RIBC2 groups. Using these signatures, we will select candidate drugs via network pharmacology, followed by molecular docking against potential targets and experimental validation.

Collectively, these findings establish *RIBC2* as a previously unrecognized diagnostic and prognostic biomarker in EC, potentially contributing to tumor initiation and progression by enhancing malignant cellular behaviors. Its consistent overexpression across both experimental models and clinical samples, along with its association with an immune-excluded tumor microenvironment, underscores its potential as a biomarker for early detection and a novel therapeutic target. For example, RIBC2 can serve as a biopsy marker to aid pathologists in evaluating indeterminate dysplasia or very early lesions. Beyond tissue, minimally invasive cytology (esophageal brush/balloon) could quantify RIBC2 mRNA as part of a small RT-qPCR panel to improve detection in at-risk patients. In addition, cfRNA/EV-RNA assays may offer a non-endoscopic option for high-risk screening and post-treatment surveillance. It is worth noting that the accurate diagnostic value of RIBC2 needs validation in the real clinical world across different cohorts. Further mechanistic and translational studies will be essential to elucidate the precise role of *RIBC2* in EC pathogenesis and to determine whether therapeutic targeting of *RIBC2* could help suppress tumor initiation or enhance patient responsiveness to existing therapies.

## Supporting information

**S1 Table. The basic information of EC patients in low- and high-risk groups.**
(DOCX)

**S2 Table. List of 22 genes significantly associated with the prognosis of EC patients.**
(DOCX)

**S3 Table. Correlation analysis between RIBC2 expression levels and various clinicopathological characteristics of EC patients.**
(DOCX)

**S1 Fig. Integrated transcriptomic analysis revealing common DEGs shared between the in vitro malignant transformation model and clinical EC tissue datasets.** (A, B) Volcano plots and heatmap showing DEGs between Het-1A-T and Het-1A-N cells. (C, D) Volcano plots and heatmap showing DEGs between tumor and normal tissues in the TCGA-EC cohort. (E) Venn diagram showing the overlap between DEGs identified in the in vitro model and in vivo tissues. (F) Functional enrichment analysis of the 22 prognosis-related DEGs.
(TIF)

**S2 Fig. Kaplan–Meier survival analysis of the low- and high-risk groups in the TCGA-EAC and TCGA-ESCC cohorts.**
(JPG)

**S3 Fig. Kaplan–Meier survival analysis of the low- and high-RIBC2 expression groups in the TCGA-EAC and TCGA-ESCC cohorts.**
(JPG)

**S4 Fig. ROC curves of RIBC2 in the diagnosis of TCGA-EAC and TCGA-ESCC patients.**
(JPG)

**S5 Fig. RIBC2-associated immune context, differential expression, pathway enrichment and PPI network in EC.**
(A) Comparison of immune cell infiltration between low- and high-RIBC2 expression groups. (B) GSEA for KEGG pathways using genes ranked by the differential signal between high- and low-RIBC2 expression groups. (C) Volcano plot showing DEGs between high- and low-RIBC2 expression groups. (D) Functional enrichment analysis of DEGs between high- and low-RIBC2 expression groups. (E) PPI network highlighting RIBC2 and its candidate partners derived from STRING. Edges indicate curated or predicted interactions.
(JPG)

**S6 Fig. Validation of RIBC2 by RT-qPCR using RIBC2 primer 2.** Analysis of RIBC2 mRNA expression in 20 paired EC tissues and their corresponding adjacent normal tissues.
(JPG)

**S1 File. The original blots.**
(PDF)

## Acknowledgments

We are grateful for excellent animal care by the animal care takers of the North China University of Technology, Hebei, China. We thank the sharing of sequencing data of EC in TCGA and GEO databases.

## Author contributions

**Conceptualization:** Guogui Sun.

**Data curation:** Xuan Zheng, Xuemin Yao.

**Formal analysis:** Xuan Zheng, Yishuang Cui, Yanan Wu.

**Investigation:** Weinan Yao.

**Methodology:** Xuan Zheng, Yishuang Cui.

**Project administration:** Guogui Sun.

**Resources:** Yanna Bi.

**Software:** Xuan Zheng, Yishuang Cui, Xuemin Yao, Junqing Gan.

**Supervision:** Guogui Sun.

**Validation:** Yanlei Ge, Ye Jin.

**Visualization:** Xuan Zheng.

**Writing – original draft:** Xuan Zheng.

**Writing – review & editing:** Guogui Sun.

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
