## [Decision Letter · Decision Letter 0]

16 Sep 2025

Dear Dr. Sun,

We look forward to receiving your revised manuscript.

Kind regards,

Mohammad H. Ghazimoradi

Academic Editor

PLOS ONE

Journal Requirements:

2. To comply with PLOS One submissions requirements, in your Methods section, please provide additional information regarding the experiments involving animals and ensure you have included details on (1) methods of sacrifice, (2) methods of anesthesia and/or analgesia, and (3) efforts to alleviate suffering.

[This research was funded by the National Natural Science Foundation of China, grant number 82472636; Hebei Province Innovation Capability Enhancement Plan Project, grant number 235A2403D; Hebei Provincial Department of Education Hebei Experimental Teaching and Teaching Laboratory Construction Project, grant number 81 and High level Research and Innovation Team Construction Plan of School of Public Health, North China University of Science and Technology, grant number KYTD202309.].

4. Thank you for stating the following in your manuscript:

[This research was funded by the National Natural Science Foundation of China, grant number 82472636; Hebei Province Innovation Capability Enhancement Plan Project, grant number 235A2403D; Hebei Provincial Department of Education Hebei Experimental Teaching and Teaching Laboratory Construction Project, grant number 81 and High level Research and Innovation Team Construction Plan of School of Public Health, North China University of Science and Technology, grant number KYTD202309.]

[This research was funded by the National Natural Science Foundation of China, grant number 82472636; Hebei Province Innovation Capability Enhancement Plan Project, grant number 235A2403D; Hebei Provincial Department of Education Hebei Experimental Teaching and Teaching Laboratory Construction Project, grant number 81 and High level Research and Innovation Team Construction Plan of School of Public Health, North China University of Science and Technology, grant number KYTD202309.]

6. Please upload a new copy of Figure S1 as the detail is not clear. Please follow the link for more information: https://blogs.plos.org/plos/2019/06/looking-good-tips-for-creating-your-plos-figures-graphics/

Reviewers' comments:

Reviewer's Responses to Questions

**Comments to the Author**

1. Is the manuscript technically sound, and do the data support the conclusions?

Reviewer #1: Yes

Reviewer #2: No

2. Has the statistical analysis been performed appropriately and rigorously?

Reviewer #1: Yes

Reviewer #2: Yes

3. Have the authors made all data underlying the findings in their manuscript fully available?

Reviewer #1: Yes

Reviewer #2: Yes

4. Is the manuscript presented in an intelligible fashion and written in standard English?

Reviewer #1: No

Reviewer #2: Yes

Reviewer #1: This manuscript establishes an MNNG-induced malignant transformation model of esophageal epithelial cells and combines transcriptomic sequencing with machine learning methods to identify and validate RIBC2 as a novel biomarker for esophageal cancer diagnosis and prognosis. The authors employed a systematic research strategy, starting from an in vitro malignant transformation model and integrating transcriptomic data from TCGA and GEO databases. They utilized multiple machine learning algorithms to construct diagnostic models and validated the biological significance of RIBC2 through clinical samples and functional experiments. The study found that RIBC2 is significantly overexpressed in transformed cells and esophageal cancer tissues, correlates with poor patient prognosis, and may influence immunotherapy response by regulating the immune microenvironment. However, the manuscript still requires further improvement and refinement in several aspects.

Major Comments

1. The authors should better elaborate in the introduction section on the important role of current multi-omics technologies in cancer biomarker discovery, particularly the application prospects of machine learning and bioinformatics methods in esophageal cancer research. With the rapid development of high-throughput sequencing and multi-omics technologies, bioinformatics has become an important tool for cancer biomarker discovery. Machine learning methods show tremendous potential in improving diagnostic accuracy and prognostic prediction, while research on immune checkpoint inhibitor-related immune toxicity also provides new perspectives for understanding the tumor immune microenvironment (doi:10.1002/imt2.70070,10.1002/mdr2.70007, PMID:37904563, 39323625).

2. The authors should use the IOBR package and GseaVis package for further analysis of immune infiltration and biological mechanisms (doi: 10.1002/mdr2.70000; 10.1002/mdr2.70001).

3. The authors should supplement more experimental evidence regarding the functional mechanisms of RIBC2. Although the study demonstrates the effects of RIBC2 knockdown on cellular phenotypes, it lacks in-depth exploration of its specific molecular mechanisms. It is recommended to add bioinformatics analyses such as pathway enrichment analysis and protein-protein interaction network analysis, as well as related molecular mechanism validation experiments.

4. The authors could more deeply explore the role of RIBC2 in other cancer types and its potential value as a therapeutic target in the discussion section. The current discussion is relatively simple and lacks in-depth analysis of RIBC2's broad biological functions.

5. The authors should discuss in more detail the importance of multi-omics technologies in future drug development to provide a theoretical foundation for developing RIBC2 as a prognostic marker (doi: 10.1016/j.cpan.2024.12.002, 10.1016/j.cpan.2024.12.001).

Minor Comments

1. The authors should carefully check the full names preceding abbreviations throughout the text to ensure that all abbreviations have their full forms when first mentioned.

2. The article requires professional native-speaker editing to reduce grammatical errors and improve idiomatic expressions.

Reviewer #2: The authors concluded that RIBC2 as a novel driver of EC initiation and progression, offering promise for early detection and therapeutic targeting. This manuscript seems to be informative for the readers of this journal if their constatation is scientifically true. There are many issues to be clarified before acceptance in this manuscript.

1. Please cite the source for the first sentence of the introduction. Based on Global Cancer Observatory, Esophageal cancer is the eleventh most commonly occurring cancer worldwide and it was ranked seventh in overall mortality rate in 2022.

2. The histological type of the Esophageal cancer cell lines should be indicated. KYSE-410 and TE-10 are SCC, and OE33 is adenocarcinoma. The most significant issue with this paper is that it analyzes SCC and adenocarcinoma together. These two histological types differ significantly in molecular oncology, so they should be considered separately even in basic experiments. At the very least, this point should be addressed in the discussion.

3. In RT-qPCR or western blot, it has long been argued that GAPDH is not suitable as a reference gene due to its excessive number of isoforms and high expression levels. Please comment on this.

4. In Fig.1-J, background stain of IHC of Pan-CK and Ki-67 are both too strong. I must say the quality of IHC is poor.

5. In Fig.3-C, is there any information of these TCGA-EC cohort 161 Esophageal cancer patients? How many SCC and adenocarcinoma ?

6. If, as the author states, this marker is useful for the early diagnosis of esophageal cancer, then the discussion should address specifically how it could potentially be utilized.

**Do you want your identity to be public for this peer review?** For information about this choice, including consent withdrawal, please see our Privacy Policy

Reviewer #1: No

Reviewer #2: **Yes:**  Yusuke Sato

---

## [Author Response · Author response to Decision Letter 1]

26 Oct 2025

Dear Editor:

We thank the Academic Editor and Reviewers for their thoughtful and constructive comments, which substantially improved our work. Following the suggestions, we have revised the manuscript and supporting files accordingly. Below we provide a point-by-point response, indicating where changes were made in the revised manuscript with line references.

#Response to Journal Requirements

Thank you for the comment. We have read the requirements carefully and revised the manuscript to conform to PLOS ONE style.

2. To comply with PLOS One submissions requirements, in your Methods section, please provide additional information regarding the experiments involving animals and ensure you have included details on (1) methods of sacrifice, (2) methods of anesthesia and/or analgesia, and (3) efforts to alleviate suffering.

Thank you for the comment. In accordance with PLOS ONE policies and the ARRIVE guidelines, we have revised the materials and methods in the revised manuscript (Lines 307-315) as “At the end of the experiment, mice were euthanized by cervical dislocation under deep anesthesia induced with sodium pentobarbital (40 mg·kg⁻¹, i.p.). Anesthetic depth was verified by the absence of pedal and palpebral reflexes. Death was confirmed by the absence of heartbeat and corneal reflex for ≥2 min. Tumors were then excised and weighed for further analysis. All animal experiments were carried out with approved protocols, relevant guidelines and regulations of the Laboratory Animal Ethics Committee of North China University of Technology (protocol code: 2023-SY-070). All methods were reported in accordance with ARRIVE guidelines.”.

[This research was funded by the National Natural Science Foundation of China, grant number 82472636; Hebei Province Innovation Capability Enhancement Plan Project, grant number 235A2403D; Hebei Provincial Department of Education Hebei Experimental Teaching and Teaching Laboratory Construction Project, grant number 81 and High level Research and Innovation Team Construction Plan of School of Public Health, North China University of Science and Technology, grant number KYTD202309.].

Thank you for the guidance. We have added a “Role of the funders” statement. The funders had no role in study design, data collection and analysis, decision to publish, or preparation of the manuscript. The cover letter has been updated accordingly.

4. Thank you for stating the following in your manuscript:

[This research was funded by the National Natural Science Foundation of China, grant number 82472636; Hebei Province Innovation Capability Enhancement Plan Project, grant number 235A2403D; Hebei Provincial Department of Education Hebei Experimental Teaching and Teaching Laboratory Construction Project, grant number 81 and High level Research and Innovation Team Construction Plan of School of Public Health, North China University of Science and Technology, grant number KYTD202309.]

[This research was funded by the National Natural Science Foundation of China, grant number 82472636; Hebei Province Innovation Capability Enhancement Plan Project, grant number 235A2403D; Hebei Provincial Department of Education Hebei Experimental Teaching and Teaching Laboratory Construction Project, grant number 81 and High level Research and Innovation Team Construction Plan of School of Public Health, North China University of Science and Technology, grant number KYTD202309.]

Thank you for the clarification. We have removed all funding-related text from the Acknowledgments and other sections of the manuscript. We will keep funding information solely in the Funding Statement per PLOS ONE policy.

Thank you for the guidance. We confirm that our revised submission fully adheres to PLOS ONE blot/gel reporting requirements. We have provided the original, uncropped and unadjusted images underlying all blots, compiled as a single Supporting Information file named as “S1_File”. Also, we have included the information of original blots in the cover letter accordingly.

Thank you for the clarification. We have removed the ethics statement from all sections other than materials and methods (Lines 113-117 and 312-315 in the revised manuscript) as “All experiments involving human tissues and clinical information were approved by the Ethics Committee of Tangshan People’s Hospital (Approval No.: RMYY-LLKS-2024042). All experiments were performed in accordance with the Declaration of Helsinki. Informed consents were obtained from all participants in the current study.” and “All animal experiments were carried out with approved protocols, relevant guidelines and regulations of the Laboratory Animal Ethics Committee of North China University of Technology (protocol code: 2023-SY-070). All methods were reported in accordance with ARRIVE guidelines.”, in accordance with PLOS ONE requirements.

6. Please upload a new copy of Figure S1 as the detail is not clear. Please follow the link for more information: https://blogs.plos.org/plos/2019/06/looking-good-tips-for-creating-your-plos-figures-graphics/

Thank you for pointing this out. We have reorganized the Figure S1 following PLOS ONE figure guidelines as shown below.

S1 Fig. Integrated transcriptomic analysis revealing common DEGs shared between the in vitro malignant transformation model and clinical EC tissue datasets. (A, B) Volcano plots and heatmap showing DEGs between Het-1A-T and Het-1A-N cells. (C, D) Volcano plots and heatmap showing DEGs between tumor and normal tissues in the TCGA-EC cohort. (E) Venn diagram showing the overlap between DEGs identified in the in vitro model and in vivo tissues. (F) Functional enrichment analysis of the 22 prognosis-related DEGs.

#Reviewer 1

This manuscript establishes an MNNG-induced malignant transformation model of esophageal epithelial cells and combines transcriptomic sequencing with machine learning methods to identify and validate RIBC2 as a novel biomarker for esophageal cancer diagnosis and prognosis. The authors employed a systematic research strategy, starting from an in vitro malignant transformation model and integrating transcriptomic data from TCGA and GEO databases. They utilized multiple machine learning algorithms to construct diagnostic models and validated the biological significance of RIBC2 through clinical samples and functional experiments. The study found that RIBC2 is significantly overexpressed in transformed cells and esophageal cancer tissues, correlates with poor patient prognosis, and may influence immunotherapy response by regulating the immune microenvironment. However, the manuscript still requires further improvement and refinement in several aspects.

1. The authors should better elaborate in the introduction section on the important role of current multi-omics technologies in cancer biomarker discovery, particularly the application prospects of machine learning and bioinformatics methods in esophageal cancer research. With the rapid development of high-throughput sequencing and multi-omics technologies, bioinformatics has become an important tool for cancer biomarker discovery. Machine learning methods show tremendous potential in improving diagnostic accuracy and prognostic prediction, while research on immune checkpoint inhibitor-related immune toxicity also provides new perspectives for understanding the tumor immune microenvironment (doi:10.1002/imt2.70070,10.1002/mdr2.70007, PMID:37904563, 39323625).

Thank you for the suggestion. Accordingly, we have expanded the introduction to outline the important role of multi-omics in cancer biomarker discovery, described the application prospects of machine learning and bioinformatics in the diagnosis, prognosis, therapeutic response and tumor microenvironment of EC as “With the advent of high-throughput sequencing and integrative multi-omics technologies, bioinformatics has emerged as a powerful tool for cancer biomarker discovery [3,4]. Through the systematic analysis of large-scale datasets, bioinformatic approaches enable the identification of critical molecular alterations and signaling pathways involved in tumor initiation and progression. In the case of EC, computational analyses have proven valuable for prioritizing candidate biomarkers with diagnostic, prognostic, and therapeutic potential. For instance, Ren et al. utilized transcriptomic data from the TCGA-EC cohort to identify ANGPTL7, CSRP1, EXPH5, F2RL2, KCNMA1, MAGEC3, MAMDC2, and SLC4A9 as fibroblast-associated gene signatures linked to EC prognosis [5]. Similarly, Yuan et al. analyzed TCGA gene expression profiles and identified miR-205-3p, miR-452-3p, and miR-6499-3p as promising biomarkers for EC staging [6]. By high-dimensional single-cell proteomics, Han et al. built a CCR4/CCR6 chemokine-based model to stratify EC patients with different response to neoadjuvant chemoradiotherapy combined with immunotherapy [7]. The microbiomes of EC patients showed that intratumoral Streptococcus signatures could predict the response of neoadjuvant chemoradiotherapy [8]. Additionally, integrative multi-omics, combining transcriptomics, proteomics, metabolomics and microbiomics may provide complementary signals and better understanding of molecular mechanisms [9]. In EC, for example, PGK1 is found to reprogram glucose metabolism and contributes to EC progression by analyzing data of genomics, proteomics and phosphoproteomics from EC patients covering 9 histopathological stages and 3 phases [10]. Integrated analyses of genomics, epigenomics, transcriptomics and proteomics could classify EC patients into four molecular subtypes and identify 28 features in the prediction of anti-PD1 therapy response [11].

Machine learning provides practical tools for feature selection and model construction when analyzing multi-omics data, and may improve the diagnostic accuracy and prognostic prediction in EC. For example, HSPD1 and MAP1LC3B were identified as key prognostic genes by LASSO regression, random forest and XGBoost algorithms [12]. Li et al. established a clinic-radiomics nomogram by machine learning to predict the overall survival after definitive chemotherapy of EC patients [13]. Sun et al. employed 10 machine learning algorithms to generate 101 algorithm combinations and identified key neoantigen-related genes. These genes could stratify EC patients into groups with differential infiltration levels of dendritic cells, macrophages and B cells [14]. However, the strength of bioinformatics predictions is often constrained by the lack of clinical and experimental validation. Therefore, transforming in silico findings into clinically meaningful insights demands rigorous analytical frameworks coupled with comprehensive biological verification.” (Lines 67-90 in the revised manuscript). These relevant citations have been added accordingly (Lines 621-645 in the revised manuscript).

2. The authors should use the IOBR package and GseaVis package for further analysis of immune infiltration and biological mechanisms (doi: 10.1002/mdr2.70000; 10.1002/mdr2.70001).

Thank you for the suggestion. In the revision, we incorporated the IOBR R package for immune deconvolution and tumor microenvironment (TME) scoring, and used GseaVis for enhanced visualization of GSEA/GSVA results to further investigate the potential immune-related mechanisms of RIBC2. Specifically, we (i) estimated immune-cell fractions with multiple algorithms (CIBERSORT, EPIC, quanTIseq, xCell, MCP-counter) implemented in IOBR, (ii) performed pathway enrichment by GseaVis package for key pathways in the high- and low-RIBC2 expression groups. We found that the infiltration of CD4 Tem and CD8 T cells were significantly higher in the low RIBC2 expression group, while the abundance of Th2 cells were much higher in the high RIBC2 expression group. Further GSEA analysis showed that pyrimidine metabolism, cell cycle, mismatch repair and DNA replication were activated, while O-glycan biosynthesis and sphingolipid metabolism were suppressed in the high RIBC2 expression group as shown below. Accordingly, we have updated the methods as “In addition, to investigate the role of RIBC2 in tumor immune microenvironment and its potential molecular mechanisms in regulating EC, we (1) estimated the immune infiltration by IOBR package [15], (2) performed GSEA by GseaVis package [16], (3) identified the DEGs by DESeq2 using |log₂ fold change| > 1 and an adjusted p-value < 0.05, and performed functional enrichment and protein-protein interaction network in low- and high-RIBC2 expression groups.” (Lines 184-189) and results as “We also found that the infiltration of CD4 Tem and CD8 T cells were significantly higher in the low RIBC2 expression group, while the abundance of Th2 cells were much higher in the high RIBC2 expression group (Fig S2A). Further GSEA analysis showed that pyrimidine metabolism, cell cycle, mismatch repair and DNA replication were activated, while O-glycan biosynthesis and sphingolipid metabolism were suppressed in the high RIBC2 expression group (Fig S2B). Moreover, we identified a total of 1,255 DEGs between low- and high-RIBC2 expression groups (Figure S2C). These DEGs were significantly enriched into KEGG pathways of neuroactive ligand-receptor interaction, cornified envelope formation, nicotine addition, biological processes of immune response and peptidase activity (Figure S2D). Moreover, by PPI analysis, we foun

---

## [Decision Letter · Decision Letter 1]

2 Nov 2025

Dear Dr. Sun,

Thank you for submitting your manuscript to PLOS ONE. After careful consideration, we feel that it has merit but does not fully meet PLOS ONE’s publication criteria as it currently stands. Therefore, we invite you to submit a revised version of the manuscript that addresses the points raised during the review process.

We look forward to receiving your revised manuscript.

Kind regards,

Mohammad H. Ghazimoradi

Academic Editor

PLOS ONE

Journal Requirements:

Reviewers' comments:

Reviewer's Responses to Questions

**Comments to the Author**

Reviewer #1: (No Response)

Reviewer #2: All comments have been addressed

2. Is the manuscript technically sound, and do the data support the conclusions?

Reviewer #1: (No Response)

Reviewer #2: Yes

3. Has the statistical analysis been performed appropriately and rigorously?

Reviewer #1: (No Response)

Reviewer #2: Yes

4. Have the authors made all data underlying the findings in their manuscript fully available?

Reviewer #1: (No Response)

Reviewer #2: Yes

5. Is the manuscript presented in an intelligible fashion and written in standard English?

Reviewer #1: (No Response)

Reviewer #2: Yes

Reviewer #1: The authors have satisfactorily addressed the previous queries. This version of manuscript is acceptable for the journal.

Reviewer #2: The authors appropriately revised their manuscript in response to the reviewers' comments. However, there are still several points that need to be clarified.

1. According to Primer Blast, the product length of the primer set of RIBC2 is 215 bp. Should the product length not be limited to 150 bp or less since mRNA extracted from FFPE is highly fragmented? Otherwise, RT-qPCR will be influenced by the degree of FFPE fragmentation rather than the actual mRNA expression levels. Please comment on this.

2. According to the provided information of TCGA-EC cohort 161 Esophageal cancer patients in Fig.3-C, the low-risk group has a higher incidence of ESCC, while the high-risk group has a higher incidence of EAC. Shouldn't the authors draw separate K-M curves for ESCC and EAC? The same applies to Fig.4-C.

3. Experiments shown in Figure 1, Het-1-T should be ESCC. However, the authors used OE33 (EAC) as a positive control. This experimental design has problem.

4. In Figure1-I, there is an error in the notation of the excised tumor. First line is OE33, second line is Het-1-T and third line is Het-1-N.

**Do you want your identity to be public for this peer review?** For information about this choice, including consent withdrawal, please see our Privacy Policy

Reviewer #1: No

Reviewer #2: No

---

## [Author Response · Author response to Decision Letter 2]

25 Dec 2025

Dear Editor and Reviewers:

We thank the Reviewers for their thoughtful and constructive comments, which substantially improved our work. Following the suggestions, we have revised the manuscript and supporting files accordingly. Below we provide a point-by-point response, indicating where changes were made in the revised manuscript with line references.

#Reviewer 1

1. The authors have satisfactorily addressed the previous queries. This version of manuscript is acceptable for the journal.

We sincerely appreciate your positive evaluation of our revised manuscript and your recognition that the current version is acceptable for publication. Thank you again for your insightful comments and constructive suggestions in the previous round, which have greatly improved the quality and clarity of our work.

#Reviewer 2

1. According to Primer Blast, the product length of the primer set of RIBC2 is 215 bp. Should the product length not be limited to 150 bp or less since mRNA extracted from FFPE is highly fragmented? Otherwise, RT-qPCR will be influenced by the degree of FFPE fragmentation rather than the actual mRNA expression levels. Please comment on this.

We thank the reviewer for this insightful and technically important comment. We fully agree with the reviewer that, for RNA extracted from FFPE tissues, amplicon length should preferably be kept as short as possible (typically ~70-150bp) in order to minimise the impact of RNA fragmentation and crosslinking on RT-qPCR quantification. As the reviewer pointed out, longer amplicons require longer contiguous RNA templates and are therefore more sensitive to variability in RNA integrity, under such conditions, Ct values may partially reflect the degree of fragmentation rather than true transcript abundance.

In our original experiments, the RIBC2 primer pair indeed generated a 215 bp product, which is not optimal for FFPE-derived RNA and could potentially introduce such bias. Several factors in our design helped mitigate this risk to some extent: (1) all 50 pairs of tumour and adjacent non-tumour tissues where obtained from matched FFPE blocks and processed under the same pre-analytical conditions, thereby reducing systematic differences in RNA degradation across groups; and (2) our main analyses focused on within-pair tumour-normal comparisons rather than comparisons across unrelated specimens.

Nevertheless, we fully acknowledge that a shorter amplicon represents best practice. To directly address this concern, we redesigned the RIBC2 primers (F: 5’-AGGCCCTCTACACAGAGACAAG-3’, R: 5’-GCTTTTTCCTTTCCACTGACTC) to yield a shorter amplicon (147 bp, the primer blast result was shown below). Due to the limited amount of remaining RNA in some cases, we repeated the RT-qPCR analysis in a subset of 20 paired FFPE-derived tumour and adjacent non-tumour tissues from the original 50-pair cohort. The new primer pair produced a single specific melting peak and robust amplification curves. Importantly, the expression patterns of RIBC2 remained unchanged that RIBC2 mRNA levels were still significantly higher in tumour tissues than in matched adjacent tissues in this 20-pair subset as shown below. These results indicate that our RT-qPCR reflect genuine difference RIBC2 expression rather than an artefact driven by FFPE RNA fragmentation.

In the revised manuscript, we have (1) updated the Methods section to clarify the use of a short-amplicon RIBC2 primer pair for FFPE-derived RNA, (2) included the redesigned RIBC2 primer sequence in the Methods (Lines 207-219 in the revised manuscript), and (3) added the short-amplicon RT qPCR results from the 20 paired samples as supplementary data (Figure S6).

The primer blast results of RIBC2

The mRNA expression of RIBC2 in 20-paired tumour tissues and adjacent tissues using new the RIBC2 primer

2. According to the provided information of TCGA-EC cohort 161 Esophageal cancer patients in Fig.3-C, the low-risk group has a higher incidence of ESCC, while the high-risk group has a higher incidence of EAC. Shouldn't the authors draw separate K-M curves for ESCC and EAC? The same applies to Fig.4-C.

We appreciate the reviewer’s thoughtful comment on the histological heterogeneity within the TCGA-EC cohort and fully agree that oesophageal squamous cell carcinoma (ESCC) and oesophageal adenocarcinoma (EAC) should be carefully considered when evaluating prognostic markers.

In response, we performed additional Kaplan–Meier survival analyses stratified by histological subtype for both the risk groups defined by our 17-gene signature and for RIBC2 expression. Specifically, TCGA-EC patients were first separated into ESCC and EAC according to the TCGA annotations, and within each subtype we compared overall survival between high- and low-risk groups and between high- and low-RIBC2 expression groups.

For the 17-gene risk signature, high-risk patients had significantly worse survival in both ESCC and EAC as shown below. For RIBC2 expression, the adverse prognostic impact of high expression was evident in EAC, whereas in ESCC the Kaplan–Meier curves did not show a statistically significant difference between the high- and low-expression groups as shown below. Meanwhile, we also draw separate ROC curves of RIBC2 for EAC and ESCC, and found that the AUC values were 0.772 in the EAC and 0.812 in the ESCC. These findings suggest that RIBC2 is involved in the development and progression of EAC and ESCC.

To avoid over-complicating the main figures, we have included the subtype-stratified Kaplan–Meier curves as new supplementary figures (Fig. S2 for the risk score and Fig. S3 for RIBC2 expression) and ROC curves of RIBC2 for EAC and ESCC (Fig. S4) and have revised the Results (Lines 418-423 and 452-458 in the revised manuscript).

Kaplan–Meier survival analysis of the low- and high-risk groups in the TCGA-EAC and TCGA-ESCC cohorts.

Kaplan–Meier survival analysis of the low- and high-RIBC2 expression groups in the TCGA-EAC and TCGA-ESCC cohorts.

ROC curves showing the diagnostic potential of RIBC2 in the TCGA-EAC and TCGA-ESCC cohorts

3. Experiments shown in Figure 1, Het-1-T should be ESCC. However, the authors used OE33 (EAC) as a positive control. This experimental design has problem.

We appreciate the reviewer’s insightful comment. As the reviewer correctly points out, Het-1A is an immortalised human oesophageal squamous epithelial cell line, and its transformed counterpart Het-1A-T is expected to resemble ESCC rather than EAC. From the perspective of histological consistency, an ESCC cell line is therefore more appropriate as a positive control for the transformation and tumourigenicity experiments.

In our original experimental design, we indeed included the ESCC cell line KYSE-450 in the subcutaneous xenograft assays together with the EAC cell line OE33 and Het-1A-T. KYSE-450 and OE33 were used as ESCC and EAC positive controls, respectively, whereas Het-1A-N served as a negative control. These experiments consistently showed that Het-1A-T formed tumours in nude mice, with tumour growth and malignant histological features that were clearly more aggressive than Het-1A-N and comparable to established oesophageal cancer cell lines. However, in the previous version of the figure, the layout and legend did not sufficiently emphasise KYSE-450 as the ESCC positive control, which may have given the impression that our design relied mainly on OE33 as the reference.

To clarify this point and to better reflect the histological context, we have reorganised Figure 1 in the revised manuscript. The panels now explicitly show four groups in the subcutaneous xenograft model: KYSE-450 (ESCC positive control), OE33 (EAC positive control), Het-1A-T (transformed squamous epithelial cells) and Het-1A-N (negative control). The figure legend and the corresponding text in the Results section have been revised to clearly state the role of KYSE-450 as the ESCC positive control (Lines 376-381 and Lines 807-816 in the revised manuscript). These revisions make our experimental design more transparent and support the conclusion that Het-1A-T recapitulates key features of squamous epithelial malignant transformation and tumourigenicity in vivo.

Figure 1. Establishment of an in vitro model of esophageal epithelial malignant transformation. (A) Representative H&E-stained images showing distinct morphological differences between non-transformed Het-1A-N and transformed Het-1A-T cells (magnification, ×100). (B, C) Cell growth curves (B) and doubling time (C) of Het-1A-N and Het-1A-T cells. (D-G) Evaluation of malignant characteristics in Het-1A-N and Het-1A-T cells using soft agar colony formation (D, ×40), colony formation (E), wound healing (F, ×40), and Transwell invasion (G, ×100) assays. (H) Images of nude mice bearing subcutaneous xenografts generated from KYSE-450 (ESCC positive control), OE33 (EAC positive control), Het-1A-T (transformed squamous epithelial cells) and Het-1A-N (negative control). (I) Photographs of resected xenograft tumours from KYSE-450, OE33 and Het-1A-T groups; no palpable tumours were formed by Het-1A-N cells. (J) Representative H&E staining of xenograft tumours derived from KYSE-450, OE33 and Het-1A-T cells; higher-magnification views of the boxed areas are shown in the lower panels. (K) Representative immunohistochemical staining of xenograft tumors showing Pan-CK and Ki-67 expression (magnification, ×200). *P < 0.05, **P < 0.01 and ****P < 0.0001.

4. In Figure1-I, there is an error in the notation of the excised tumor. First line is OE33, second line is Het-1-T and third line is Het-1-N.

We thank the reviewer for carefully checking the xenograft images. In our xenograft experiments, only KYSE-450, OE33 and Het-1A-T formed palpable tumours, whereas Het-1A-N did not generate any measurable masses in nude mice. In the revised manuscript, we have corrected Figure 1I so that the excised tumours are properly labelled according to their cell-line origin (KYSE-450, OE33 and Het-1A-T). The figure legend and the corresponding text in the Results section have been updated (Lines 376-381 and Lines 807-816 in the revised manuscript) accordingly to clarify that Het-1A-N cells failed to form tumours in vivo.

---

## [Decision Letter · Decision Letter 2]

29 Dec 2025

Integrative bioinformatics and experiments identify RIBC2 as a key regulator in the esophageal cancer

PONE-D-25-45730R2

Dear Dr. Sun,

We’re pleased to inform you that your manuscript has been judged scientifically suitable for publication and will be formally accepted for publication once it meets all outstanding technical requirements.

Kind regards,

Mohammad H. Ghazimoradi

Academic Editor

PLOS One

Additional Editor Comments (optional):

Reviewers' comments:

Reviewer's Responses to Questions

**Comments to the Author**

Reviewer #2: All comments have been addressed

2. Is the manuscript technically sound, and do the data support the conclusions?

Reviewer #2: Yes

3. Has the statistical analysis been performed appropriately and rigorously?

Reviewer #2: Yes

4. Have the authors made all data underlying the findings in their manuscript fully available?

Reviewer #2: Yes

5. Is the manuscript presented in an intelligible fashion and written in standard English?

Reviewer #2: Yes

Reviewer #2: The authors have satisfactorily addressed the previous issues. This version of manuscript is acceptable for the journal.

**Do you want your identity to be public for this peer review?** For information about this choice, including consent withdrawal, please see our Privacy Policy

Reviewer #2: **Yes:**  Yusuke Sato

---

## [Editor Report · Acceptance letter]

PONE-D-25-45730R2

PLOS One

Dear Dr. Sun,

I'm pleased to inform you that your manuscript has been deemed suitable for publication in PLOS One. Congratulations! Your manuscript is now being handed over to our production team.

Kind regards,

on behalf of

Dr. Mohammad H. Ghazimoradi

Academic Editor

PLOS One